# AGILE3D: ATTENTION GUIDED INTERACTIVE MULTI-OBJECT 3D SEGMENTATION

**Yuanwen Yue**[1,2]    **Sabarinath Mahadevan**[3]    **Jonas Schult**[3]    **Francis Engelmann**[2,4]
**Bastian Leibe**[3]    **Konrad Schindler**[1,2]    **Theodora Kontogianni**[2]

[1]Photogrammetry and Remote Sensing, ETH Zurich    [2]ETH AI Center, ETH Zurich
[3]Computer Vision Group, RWTH Aachen University    [4]Google

## ABSTRACT

During interactive segmentation, a model and a user work together to delineate objects of interest in a 3D point cloud. In an iterative process, the model assigns each data point to an object (or the background), while the user corrects errors in the resulting segmentation and feeds them back into the model. The current best practice formulates the problem as binary classification and segments objects one at a time. The model expects the user to provide *positive clicks* to indicate regions wrongly assigned to the background and *negative clicks* on regions wrongly assigned to the object. Sequentially visiting objects is wasteful since it disregards synergies between objects: a positive click for a given object can, by definition, serve as a negative click for nearby objects. Moreover, a direct competition between adjacent objects can speed up the identification of their common boundary. We introduce AGILE3D, an efficient, attention-based model that (1) supports simultaneous segmentation of multiple 3D objects, (2) yields more accurate segmentation masks with fewer user clicks, and (3) offers faster inference. Our core idea is to encode user clicks as spatial-temporal queries and enable explicit interactions between click queries as well as between them and the 3D scene through a click attention module. Every time new clicks are added, we only need to run a lightweight decoder that produces updated segmentation masks. In experiments with four different 3D point cloud datasets, AGILE3D sets a new state-of-the-art. Moreover, we also verify its practicality in real-world setups with real user studies. Project page: https://ywyue.github.io/AGILE3D.

## 1 INTRODUCTION

Accurate 3D instance segmentation is a crucial task for a variety of applications in computer vision and robotics. Manually annotating ground-truth segmentation masks on 3D scenes is expensive and fully-automated 3D instance segmentation approaches (Hou et al., 2019; Engelmann et al., 2020; Chen et al., 2021; Vu et al., 2022) do not generalize well to unseen object categories of an open-world setting. Interactive segmentation techniques have been commonly adopted for large-scale 2D image segmentation (Benenson et al., 2019), where a user interacts with a segmentation model by providing assistive clicks or scribbles iteratively. While interactive image segmentation has been the subject of extensive research (Xu et al., 2016; Li et al., 2018; Mahadevan et al., 2018; Jang & Kim, 2019; Kontogianni et al., 2020; Sofiiuk et al., 2022; Chen et al., 2022), there are only a few works that explore interactive 3D segmentation (Valentin et al., 2015; Shen et al., 2020; Zhi et al., 2022). Those methods either only work for semantic segmentation or require images with associated camera poses w.r.t. the 3D scene. The latter constraint makes them not suitable for non-camera sensors (*e.g.*, LiDAR), or for dynamically changing viewpoints like in AR/VR settings. Even with available images, providing feedback for the same object from multiple viewpoints can be tedious. Here, we focus on interactive segmentation directly in 3D point clouds.

Recently Kontogianni et al. (2023) introduced an approach for interactive 3D segmentation that operates directly on point clouds, achieving state-of-the-art performance. The task is formulated as a binary segmentation for one object at a time (Fig. 1, *left*), and follows the mainstream approach of interactive 2D image segmentation (Xu et al., 2016; Mahadevan et al., 2018; Li et al., 2018; Jang & Kim, 2019; Kontogianni et al., 2020; Sofiiuk et al., 2022; Chen et al., 2022): User input comes in the form of binary clicks that identify locations with incorrect labels. *Positive clicks* specify missed parts of the target object and *negative clicks* specify background areas falsely labeled as

Figure 1: **Architecture comparison.** *Left:* InterObject3D (Kontogianni et al., 2023). *Right:* our AGILE3D. Given the same set of 10 user clicks, AGILE3D can effectively segment three objects while InteObject3D can only segment one. InterObject3D takes 0.5s for one object while we need only 0.25s for all three, thanks to running a lightweight decoder per iteration and not a full forward pass through the entire network[1]. Unlike InterObject3D, our backbone learns features for all objects.

foreground. The two sets of click coordinates are encoded as binary masks and concatenated with the 3D point coordinates to form the input to a deep learning model. A subtle, but important limitation of that approach is that objects are processed sequentially, one at a time: obviously, object outlines within a scene are boundaries between different objects (rather than between an isolated object and a monolithic background): Positive clicks for one object can, by definition, serve as negative clicks for other, nearby objects. In most cases also the opposite is true, as negative clicks will often also lie on other objects of interest. Handling multiple objects jointly rather than sequentially also brings a computational advantage, because every round of user input requires a forward pass.

In this work, we propose AGILE3D, an attention-guided interactive 3D segmentation approach that can simultaneously segment multiple objects in context. Given a 3D scene, we first employ a standard 3D sparse convolutional backbone to extract per-point features *without* click input. Instead of encoding clicks as click maps, we propose to encode them as high-dimensional feature *queries* through our *click-as-query* module. To exploit not only the spatial locations of user clicks but also their temporal order in the iterative annotation process, they are supplemented by a spatial-temporal positional encoding. To enable the information exchange between different clicks, and between clicks and the 3D scene, we propose a *click attention* module, where clicks explicitly interact with the 3D scene through click-to-scene and scene-to-click attention and with each other through click-to-click attention. Finally, we aggregate the roles of all click queries to obtain a single, holistic segmentation mask in our query fusion module, and train the network in a way that regions compete for space. In this manner, AGILE3D imposes no constraint on the number of objects and seamlessly models clicks on multiple objects, including their contextual relations allowing for more accurate segmentation masks of multiple objects together. Disentangling the encoding of the 3D scene from the processing of the clicks makes it possible to pre-compute the backbone features, such that during iterative user feedback one must only run the lightweight decoder (*i.e.* click attention and query fusion), thus significantly reducing the computation time (Fig. 1). Moreover, our backbone learns the representation of all scene objects while InterObject3D only learns the representation of a single object depending on the input clicks (*c.f.* learned features converted to RGB with PCA in Fig. 1).

Our proposed method aims to be trainable with limited data, and able to compute valid segmentation masks for zero-shot and few-shot setups on unseen datasets (*e.g.*, dataset annotation). We train on a single dataset, ScanNetV2-Train (Dai et al., 2017), and then evaluate on ScanNetV2-Val (Inc. ScanNet20 and ScanNet40) (Dai et al., 2017), S3DIS (Armeni et al., 2016), KITTI-360 (Liao et al., 2022). For all of them, AGILE3D outperforms the state of the art. Our main contributions are:

1. We introduce the interactive multi-object 3D segmentation task to segment multiple objects concurrently with a limited number of user clicks in a 3D scene.
2. We propose AGILE3D, which is the first interactive approach that can segment multiple objects in a 3D scene, achieving state-of-the-art in both interactive multi- and single-object segmentation benchmarks.
3. We propose the setup, evaluation, and iterative training strategy for interactive multi-object segmentation on 3D scenes and conduct extensive experiments to validate the benefits of our task formulation.
4. We develop a user interface and perform real user studies to verify the effectiveness of our model and the proposed training strategy in real annotation tasks.

---

[1]Time is measured on a single TITAN RTX GPU.

## 2   RELATED WORK

**3D instance segmentation.**   Fully-supervised 3D instance segmentation is a well-researched problem with remarkable progress (Hou et al., 2019; Yi et al., 2019; Engelmann et al., 2020; Jiang et al., 2020; Vu et al., 2022; Schult et al., 2023; Sun et al., 2023). Like all fully-supervised methods, they require large amounts of annotated training data. To relieve the labeling cost, Hou et al. (2021); Chibane et al. (2022) explore weakly-supervised 3D instance segmentation by learning from weak annotations such as sparse points or bounding boxes with the cost of performance drop. For a survey on 3D segmentation, we refer the readers to Guo et al. (2020); He et al. (2021); Xiang et al. (2023). Interactive 3D segmentation differs from those methods. First, fully and weakly-supervised methods require per dataset training and are unable to generalize to classes that are not part of the training set. Moreover, they cannot incorporate additional user input to refine any inaccuracies further, whereas our method aims to produce high-quality masks using the model's ability to interact with humans.

**Interactive 3D segmentation.**   There are only a few approaches that support user input in generating 3D segmentation masks (Valentin et al., 2015; Shen et al., 2020; Zhi et al., 2022; Kontogianni et al., 2023). Valentin et al. (2015); Zhi et al. (2022) focus on online semantic labeling of 3D scenes rather than instance segmentation. Shen et al. (2020) shift the user interaction to the 2D domain but require images with known camera poses. Moreover, providing feedback for the same object in multiple viewpoints is cumbersome. The closest work to ours is InterObject3D (Kontogianni et al., 2023), an interactive 3D segmentation approach that directly operates on point clouds. However, InterObject3D can only segment single objects sequentially whereas our method can segment multiple objects simultaneously, leading to better results with less user interaction.

**Interactive image segmentation**   has been extensively studied (Xu et al., 2016; Li et al., 2018; Mahadevan et al., 2018; Jang & Kim, 2019; Benenson et al., 2019; Kontogianni et al., 2020; Chen et al., 2022; Sofiiuk et al., 2022; Liu et al., 2022; Du et al., 2023; Zhou et al., 2023; Wei et al., 2023) but all of the methods are designed to segment single objects. Only few works (Agustsson et al., 2019; Rana et al., 2023) explore interactive full-image or multi-object segmentation but specialize in the 2D domain. To the best of our knowledge, our method is the first approach that supports interactive multi-object segmentation in 3D point clouds. Moreover, although simulated evaluation is a well-established protocol in both the 2D and 3D domains, we urge the community to move beyond simulated clicks and also evaluate with real user studies.

## 3   METHOD

Consider a 3D scene $P \in \mathbb{R}^{N \times C}$, with $N$ number of 3D points and $C$ the feature dimension associated with each point. $C$ is normally set to 3 for locations $xyz$, otherwise 6 if colors $rgb$ are available.

**Interactive single-object segmentation.** Given such a scene, in interactive single-object segmentation (Kontogianni et al., 2023), the user provides a sequence of clicks, where positive clicks are considered on the desired object and negative clicks on the background. The segmentation mask is obtained through an iterative process: the model provides the user with a segmentation mask, then the user provides feedback to the model via positive/negative clicks. The model provides an updated mask given the user corrections. The process repeats until the user is satisfied with the result.

**Interactive multi-object segmentation.** We extend the above formulation to incorporate interactive multi-object scenarios. Let us assume a user wants to segment $M$ target objects in a scene. We denote user clicks as $S = (c_1, c_2, ..., c_k)_{k=1}^{K}$, where $k$ is the click index. $k$ also doubles as a timestamp indicator since the clicks come as a sequence in time. Each click $c_k$ is represented by two attributes $c_k = \{p_k, o_k\}$, where $p_k \in \mathbb{R}^3$ are the 3D coordinates and $o_k \in \{0, 1, ..., M\}$ is the region index, indicating whether the click comes from the background (when $o_k = 0$) or associated with object $o_k$ (when $o_k \geqslant 1$). $S$ is initialized as an empty sequence and iteratively extended when the user gives more clicks. Given the 3D scene $P$ and click sequence $S$, our goal is to predict a single, holistic segmentation mask $\mathcal{M} \in \{0, 1, ..., M\}^{N}$, which indicates the interest region each point belongs to. Please note we aim to ensure that each point belongs to a single segmentation mask, which is different from the interactive single-object segmentation task, where several passes of the same scene with different objects of interest might result in some points assigned to multiple segmentation masks. When $M = 1$, the above formulation matches the interactive single-object segmentation setting as in Kontogianni et al. (2023).

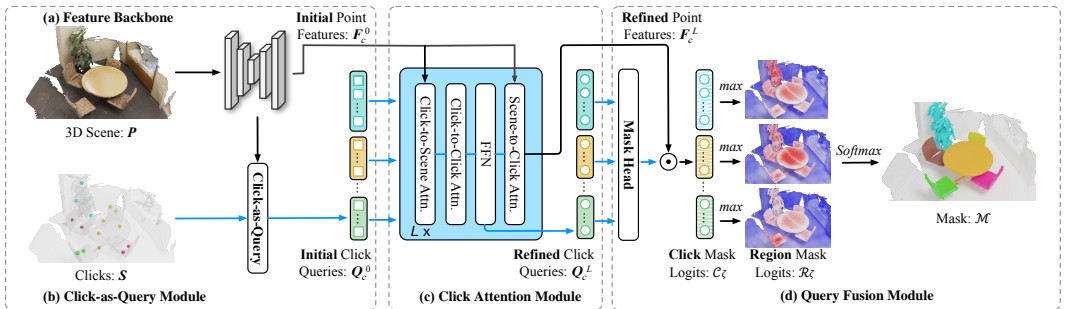

Figure 2: **Model of AGILE3D.** Given a 3D scene and a user click sequence, (a) the feature backbone extracts per-point features and (b) the click-as-query module converts user clicks to high-dimensional query vectors. (c) The click attention module refines the click queries and point features through multiple attention mechanisms. (d) The query fusion module first fuses the per-click mask logits to region-specific mask logits and then produces a final mask through a softmax. With → we denote the user click information and with → the scene information. Colors of clicks, click queries and segmentation masks are consistent for the same object.

### 3.1 AGILE3D

Our overall architecture (Fig. 2) consists of (a) a feature backbone that extracts per-point features, (b) a click-as-query module that converts user clicks to spatial-temporal query vectors, (c) a click attention module that enables explicit interaction between click queries themselves and scene features, and (d) a query fusion module that predicts a holistic segmentation of multiple objects.

**Feature backbone**. Our feature backbone is a sparse convolutional U-Net, based on the Minkowski Engine (Choy et al., 2019) as in Kontogianni et al. (2023). It takes as input a 3D scene and produces a feature map $F \in \mathbb{R}^{N' \times D}$, where $D$ is the feature dimension. Unlike Kontogianni et al. (2023) which sends the concatenation of $P$ and encoded click maps to the backbone, we only input $P$ and insert the user clicks separately in our click-as-query module.

**Click-as-query module** converts each user click $c_k$ to query $q_k$. A click is a user query indicating a desired region. The click query should properly encode representative knowledge of an object so that the system can correctly classify all relevant points. On the other hand, clicks are a sequence of 3D points that are inherently spatial and temporal. Motivated by this, we encode the click query as three parts $q_k = \{\mathbf{c}_k, \mathbf{s}_k, \mathbf{t}_k\}$, each of which models its *content*, *spatial*, and *temporal* properties. The *content* part $\mathbf{c}_k \in \mathbb{R}^D$ is initialized from the point feature map $F$ of the backbone. For each click, we find the nearest voxel position in $F$ and use the feature of that voxel as $\mathbf{c}_k$. The *spatial* part $\mathbf{s}_k \in \mathbb{R}^D$ is created by mapping the click coordinates $p_k$ to the same feature space as $\mathbf{c}_k$ using Fourier positional encodings (Tancik et al., 2020). Similarly, we transform the timestamp $k$ to a *temporal* embedding $\mathbf{t}_k \in \mathbb{R}^D$ using a sin/cos positional encoding of (Vaswani et al., 2017). We consolidate the spatial and temporal parts to a *positional* part by summing $\mathbf{s}_k$ and $\mathbf{t}_k$. After initialization, the content part of each query $\mathbf{c}_k$ along with per-point features $F$ will be iteratively refined through the click attention module by interacting with each other.

**Click attention module** is designed to enable interaction between the click queries themselves and between them and the point features. Each decoder layer consists of a click-to-scene attention module (C2S), a click-to-click attention module (C2C), a feed-forward network, and finally a scene-to-click attention module (S2C). All queries are represented by the positional and the content part. We denote the positional part of all click queries as $Q_p$ and the content part at layer $l \in \{0, 1, ..., L\}$ as $Q_c^l$. We use the same representation for the scene points. We denote the point features at layer $l$ as $F_c^l$, where $F_c^0 = F$, which represents the content part of the 3D points. The positional part of 3D points $F_p$ is encoded via Fourier positional encodings (Tancik et al., 2020) based on voxel positions to ensure access to point cloud geometric information to the decoder.

The C2S performs cross-attention from click queries to point features, which enables click queries to extract information from relevant regions in the point cloud. In the C2C, we let each click query self-attend to each other to realize inter-query communications. The C2C is followed by an FFN that further updates each query. All three steps only update click queries while the point features are static. To make the point features click-aware, we add the S2C that performs cross-attention from point features to click queries. Details on these attention formulations can be found in the Appendix.

**Query fusion module**. We apply a shared mask head (MLP) $f_{mask}(\cdot)$ to convert the per-click content embeddings $Q_c^L$ to $K$ mask embeddings and then compute the dot product between these mask embeddings and the refined point features, which produces $K$ mask logits maps $\mathcal{C}_\zeta \in \mathbb{R}^{N' \times K}$,

*i.e.*, $\mathcal{C}_\zeta = F_c^L \cdot f_{mask}(Q_c^L)^T$. Since there can be multiple queries representing the same region (object or background), we apply a per-point `max` operation on $\mathcal{C}_\zeta$ that shares the same region. This step can be achieved through the association between each click and its region index $o_k$, and gives us $M + 1$ region-specific mask logits map $\mathcal{R}_\zeta \in \mathbb{R}^{N' \times (M+1)}$. We obtain the final segmentation mask $\mathcal{M} \in \{0, 1, ..., M\}^{N'}$ through a `softmax`, which indicates the region each point belongs to.

### 3.2 USER SIMULATION AND TRAINING

Simulated clicks are commonly used in the interactive community for both training and evaluation. In Kontogianni et al. (2023) positive clicks are uniformly sampled on the object and negative ones are sampled from the object neighborhood. At test time they imitate a user who always clicks at the center of the largest error region. However, the training and test strategies are different since randomly sampled clicks are independent of network errors and lack a specific order. We propose an iterative strategy that approximates real user behavior even during training. Although iterative training has been explored in interactive image segmentation (Mahadevan et al., 2018; Sofiiuk et al., 2022), those methods can only work for single object setup. A detailed comparison with Mahadevan et al. (2018); Sofiiuk et al. (2022) can be found in Tab. 8 and the Appendix.

**Multi-object iterative training.** Our iterative strategy is shown in Fig. 3. We simulate user clicks for each batch separately in an iterative way with $n$ number of iterations sampled uniformly from 1 to $N_{iter}$. $S^i$ are the clicks sampled in the $i$-th iteration. The training starts from the initial clicks ($S^0$) collected from each target object's center. Full iterative training, like in testing, is costly, requiring an iteration after each sampled click. Therefore, when sampling clicks for the next iteration, instead of only sampling one click from the largest error region, we sample $N_i$ clicks from the top $N_i$ error regions (one click per region). This strategy can generate training samples that contain a large number of clicks in a small number of iterations, keeping the training complexity manageable. We freeze the model when sampling clicks in iterations 1 to $N_{iter} - 1$ and only allow backpropagation in the last iteration.

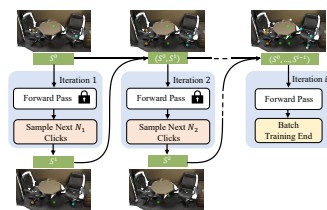

Figure 3: **Multi-object iterative training**.

**Multi-object user simulation during test.** Interactive single-object segmentation (Xu et al., 2016; Sofiiuk et al., 2022; Kontogianni et al., 2023) evaluation strategies imitate a user who clicks at the largest error region. In our multi-object scenario, we share the spirit and enable users to focus first on the largest errors in the whole scene. Our user simulation strategy starts with one click at the center of each object to get an initial prediction. We then compute a set of error clusters (comparing prediction to ground truth) and sample the next click from the center of the largest cluster.

**Loss.** We supervise our network with the cross-entropy and the Dice loss (Deng et al., 2018) for multi-class segmentation since we want neighboring masks to compete for space and ensure that each point is assigned to a single label. The number of classes varies per scene and is $M + 1$, where $M$ is the number of objects the user wants to segment. The total loss is defined as: $\mathcal{L} = \frac{1}{N} \sum_{p \in P} w_p (\lambda_{CE} \mathcal{L}_{CE}(p) + \lambda_{Dice} \mathcal{L}_{Dice}(p))$ where $\lambda_{CE}$ and $\lambda_{Dice}$ are the balancing loss weights and $w_p$ the distance of the points to the user click. Additional implementation details are in the Appendix.

## 4 EXPERIMENTS

**Tasks.** AGILE3D is a versatile model, able to perform both interactive single- and multi-object 3D segmentation. Since there is no existing benchmark for interactive multi-object segmentation we propose this new task including an evaluation protocol and metrics. We evaluate in both scenarios.

**Datasets.** A key aspect of interactive segmentation systems is their ability to work on datasets that exhibit significant variations in data distribution compared to the training data. To this end, we train the system on ScanNetV2-Train (Dai et al., 2017), an indoor dataset, and subsequently, we evaluate on datasets that follow distinct distributions, including ScanNetV2-Val (Dai et al., 2017) (same distribution), S3DIS (Armeni et al., 2016) (different sensor), and even KITTI-360 (Liao et al., 2022) (outdoor LiDAR point clouds). We also train two models on ScanNet-train: one with 40 classes and another with only the subset of 20 benchmark classes.

Table 1: **Quantitative results on interactive single-object segmentation.** We compare our method with the current state-of-the-art InterObject3D (Kontogianni et al., 2023) and our enhanced InterObject3D++ in several datasets. Our method offers 3D masks of higher quality with fewer user clicks and generalizes better to new classes and datasets.

| Method | Train → Eval | IoU@5 ↑ | IoU@10 ↑ | IoU@15 ↑ | NoC@80 ↓ | NoC@85 ↓ | NoC@90 ↓ |
|---|---|---|---|---|---|---|---|
| InterObject3D | | 67.6 | 77.6 | 81.2 | 9.6 | 11.8 | 14.6 |
| InterObject3D++ | ScanNet20 → ScanNet40 (+ *unseen*) | 76.4 | 82.2 | 83.8 | 7.8 | 10.2 | 13.4 |
| **AGILE3D (Ours)** | | **78.5** | **82.9** | **84.5** | **7.4** | **9.8** | **13.1** |
| InterObject3D | | 72.4 | 79.9 | 82.4 | 8.9 | 11.2 | 14.2 |
| InterObject3D++ | ScanNet40 → ScanNet40 | 78.0 | 82.9 | 84.2 | 7.7 | 10.0 | 13.2 |
| **AGILE3D (Ours)** | | **79.9** | **83.7** | **85.0** | **7.1** | **9.6** | **12.9** |
| InterObject3D | | 72.4 | 83.6 | 88.3 | 6.8 | 8.4 | 11.0 |
| InterObject3D++ | ScanNet40 → S3DIS-A5 | 80.8 | **89.2** | **91.5** | 5.2 | 6.7 | **9.3** |
| **AGILE3D (Ours)** | | **83.5** | 88.2 | 89.5 | **4.8** | **6.4** | 9.5 |
| InterObject3D | | 14.3 | 26.3 | 35.0 | 19.1 | 19.4 | 19.7 |
| InterObject3D++ | ScanNet40 → KITTI-360 | 19.9 | 40.6 | **55.1** | 17.0 | 17.7 | 18.4 |
| **AGILE3D (Ours)** | | **44.4** | **49.6** | 54.9 | **14.2** | **15.5** | **16.8** |

**Evaluation metrics.** For *single-object* evaluation we follow the evaluation protocol of Kontogianni et al. (2023). We compare the methods on (1) NoC@q% ↓, the average number of clicks needed to reach q% IoU, and (2) IoU@k ↑, the average IoU for k number of clicks per object (capped at 20). We extend the evaluation protocol for *multi-object*. We no more enforce a per-object budget but we allow a total budget of $M \times 20$ clicks for a user who wants to segment $M$ objects in a scene. We propose the $\overline{\text{IoU}}@\overline{k}$ ↑ metric, which represents the average IoU of all target objects after an average of k clicks allocated to each object. $\overline{k}$ is an average of k over $M$ and each object does not necessarily share exactly $\overline{k}$ clicks. Similarly, we report $\overline{\text{NoC}}@\overline{q\%}$ ↓, which represents the average number of clicks to reach an average q% IoU for all target objects in the scene.

**Baseline.** We use InterObject3D (Kontogianni et al., 2023) as our baseline in interactive single-object segmentation. However, InterObject3D is designed to segment objects sequentially and cannot directly be evaluated by our interactive multi-object segmentation protocol. To this end, we use InterObject3D to process each object with one click per object sequentially and obtain a binary mask for each object. We merge those binary masks manually and then use our user simulation protocol to sample the next click. Note this hand-crafted baseline is just for a complete comparison and cannot be seen as interactive multi-object segmentation. We additionally created a strong implementation of InterObject3D, an **enhanced** baseline (InterObject3D++) by incorporating our proposed iterative training. Outperforming this even stronger baseline shows that there is merit in leveraging the clicks of multiple objects together rather than handling objects in isolation.

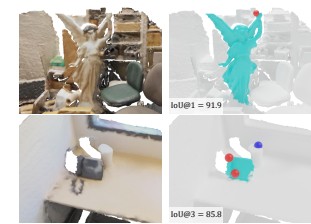

Figure 4: **Open-world segmentation** from ScanNet20. AGILE3D can segment new objects like statue and phone.

### 4.1 EVALUATION ON SINGLE-OBJECT SEGMENTATION.

**Comparison with state-of-the-art.** Results are summarized in Tables 1, 2, 3 and Fig 5, 4. We perform the evaluation in scenarios of increasing difficulty and distribution shift:

Table 2: **Quantitative results on interactive single-object segmentation (≤ 3 clicks).** Our method performs significantly better than baselines, especially in the low click regime. Our results for just 1 click are comparable or better than the 3 clicks of the baselines. Models are trained on ScanNet40.

| | Method | IoU@1 ↑ | IoU@2 ↑ | IoU@3 ↑ |
|---|---|---|---|---|
| ScanNet | InterObject3D | 40.8 | 55.9 | 63.9 |
| | InterObject3D++ | 40.7 | 60.7 | 70.2 |
| | **AGILE3D (Ours)** | **63.3** | **70.9** | **75.4** |
| S3DIS | InterObject3D | 38.5 | 54.0 | 62.5 |
| | InterObject3D++ | 32.7 | 55.8 | 69.0 |
| | **AGILE3D (Ours)** | **58.5** | **70.7** | **77.4** |
| KITTI-360 | InterObject3D | 2.0 | 5.1 | 8.5 |
| | InterObject3D++ | 3.4 | 7.0 | 11.0 |
| | **AGILE3D (Ours)** | **34.8** | **40.7** | **42.7** |

Table 3: **Comparison with fully-supervised.** We compare our method with the state-of-art fully supervised instance segmentation method. Both methods were trained on ScanNet20-seen and evaluated on the ScanNet20-seen and ScanNet20-unseen.

| | Method | #clicks | AP | AP$_{50\%}$ | AP$_{25\%}$ |
|---|---|---|---|---|---|
| Benchmark Classes | Mask3D | – | 51.5 | 77.0 | 90.2 |
| | | 1 | 53.5 | 75.6 | 91.3 |
| | | 2 | 64.0 | 86.4 | 96.0 |
| | **AGILE3D** | 3 | 70.3 | 91.4 | 98.1 |
| | (Ours) | 10 | 83.2 | 98.3 | 99.8 |
| | | 20 | 86.8 | 99.2 | 100.0 |
| Unseen Classes | Mask3D | – | 5.3 | 13.1 | 24.7 |
| | | 1 | 24.8 | 45.7 | 72.4 |
| | | 2 | 36.9 | 63.5 | 85.8 |
| | **AGILE3D** | 3 | 45.5 | 74.4 | 92.2 |
| | (Ours) | 10 | 67.8 | 94.8 | 99.7 |
| | | 20 | 74.5 | 97.6 | 99.9 |

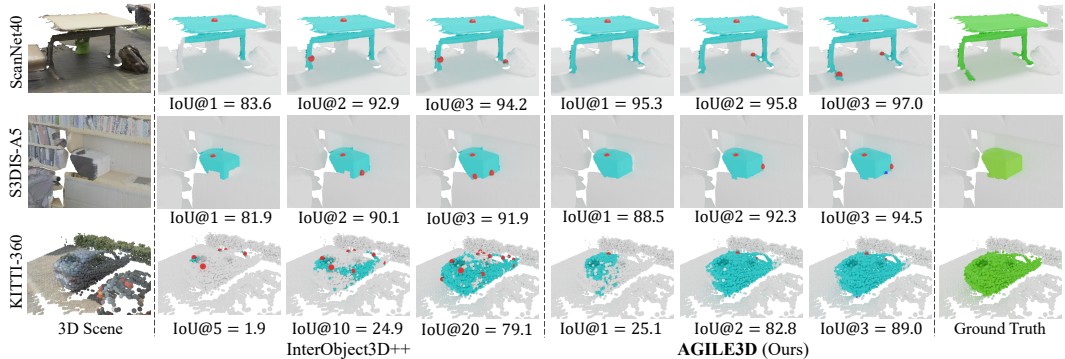

Figure 5: **Qualitative results on interactive single-object segmentation.**

*ScanNet-train→ScanNet-val*: We evaluate on the ScanNet-val dataset, considering small distribution shifts. In both the ScanNet20 and ScanNet40 setups, our method surpasses the baseline and the enhanced baseline, as shown in Tab. 1 (ln.1,2). We test the generalization of our method to novel classes by evaluating the trained model on ScanNet 20 classes with the additional 20 unseen classes. Tab. 1 clearly demonstrates that our AGILE3D outperforms the current state-of-the-art in all metrics.

*ScanNet-train→S3DIS*: We further assess the effectiveness of our model's generalization by evaluating it on S3DIS, an another indoor dataset with different characteristics than ScanNet. Our model outperforms the state-of-the-art baseline. For example, with only 5 clicks, our AGILE3D achieves an impressive IoU of 83.5 on S3DIS, surpassing the baseline's performance of 72.4.

*ScanNet-train→KITTI-360*: Our method also excels in the challenging domain shift of training on ScanNet and testing on KITTI-360, an outdoor dataset captured with a LiDAR sensor. Our method surpasses the baseline by a factor of 4 and the enhanced baseline by a factor of 2 on IoU@5.

Our method's performance is particularly impressive in the low click regime ($\leq 3$ clicks). With just a single click, our method achieves $\approx 60$ IoU on both ScanNet and the unseen dataset of S3DIS (Tab. 2). It is a significant improvement over the baseline of $\approx 40$ IoU. With three clicks, our method achieves even higher IoU scores of 75.4 and 77.4 on ScanNet and S3DIS, respectively.

**Comparison with fully-supervised methods.** Results are summarized in Tab. 3 and Fig. 4. Fully supervised methods for 3D instance segmentation achieve remarkable results on tasks and data distributions similar to those encountered during training. However, we demonstrate that with minimal human feedback, we can surpass the performance of fully supervised methods, particularly in classes that were not seen during training. Our method achieves precision 4 times higher than the state-of-the-art method Mask3D (Schult et al., 2023) for unseen classes with just one click (Tab. 3). In Fig. 4, AGILE3D obtains high-quality masks of novel objects (*e.g.*, statue, phone) with few clicks.

## 4.2 EVALUATION ON MULTI-OBJECT SEGMENTATION.

Table 4: **Quantitative results on interactive multi-object segmentation.** We adapt the state-of-the-art method in interactive single-object segmentation to be evaluated by our multi-object protocol for a complete comparison (*Baseline* paragraph of Sec. 4). Note the baselines still predict binary masks for single-object and final masks must be merged manually.

| Method | Train → Eval | $\overline{IoU}@5\uparrow$ | $\overline{IoU}@10\uparrow$ | $\overline{IoU}@15\uparrow$ | $\overline{NoC}@80\downarrow$ | $\overline{NoC}@85\downarrow$ | $\overline{NoC}@90\downarrow$ |
|---|---|---|---|---|---|---|---|
| InterObject3D | | 75.1 | 80.3 | 81.6 | 10.2 | 13.5 | 16.6 |
| InterObject3D++ | ScanNet40 → ScanNet40 | 79.2 | 82.6 | 83.3 | 8.6 | 12.4 | 15.7 |
| **AGILE3D** (Ours) | | **82.3** | **85.0** | **86.0** | **6.3** | **10.0** | **14.4** |
| InterObject3D | | 76.9 | 85.0 | 87.3 | 6.8 | 8.8 | 13.5 |
| InterObject3D++ | ScanNet40 → S3DIS-A5 | 81.9 | 88.3 | 89.3 | 5.7 | 7.6 | 11.6 |
| **AGILE3D** (Ours) | | **86.3** | **88.3** | **90.3** | **3.4** | **5.7** | **9.6** |
| InterObject3D | | 10.5 | 22.1 | 31.0 | 19.8 | 19.8 | 19.9 |
| InterObject3D++ | ScanNet40 → KITTI-360 | 16.7 | 37.1 | **52.2** | 18.3 | 18.9 | 19.3 |
| **AGILE3D** (Ours) | | **40.5** | **44.3** | 48.2 | **17.4** | **18.3** | **18.8** |

Results are summarized in Tab. 4 and Fig. 6, 7. More qualitative results in Appendix. We adapted InterObject3D for our multi-object protocol for a complete comparison. We do not enforce a per-object click budget for the baselines but allow them to sample the next click from the biggest error region across all target objects. Nevertheless, the baselines are *still limited* to segmenting one object in each forward pass. AGILE3D outperforms all the baselines, requiring significantly fewer clicks

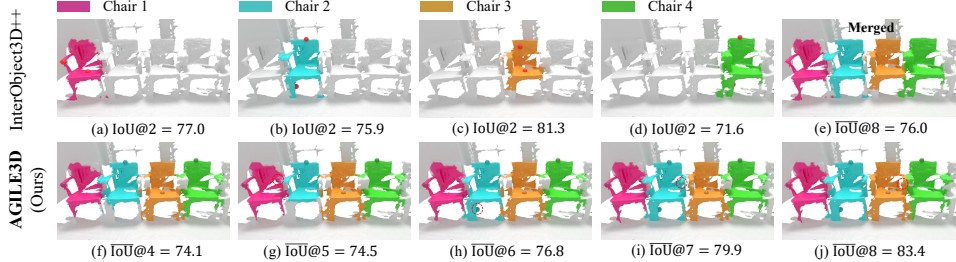

**Figure 6: Qualitative results on interactive multi-object segmentation on ScanNet40.** A user wants to segment 4 chairs using InterObject3D++ (*top*) or AGILE3D (*bottom*). (a)-(d) show the results for each object and (e) shows the final merged prediction after a total of 8 clicks for InterObject3D++. (f)-(j) show the results of AGILE3D after each iteration. New click in each iteration is marked with a red circle.

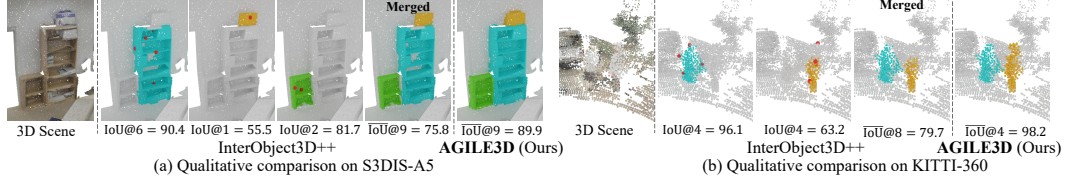

**Figure 7: More qualitative comparison.** AGILE3D achieves higher $\overline{IoU}$, *e.g.*, when segmenting cabinets and boxes on S3DIS-A5 (*left*) and pedestrians on KITTI-360 (*right*).

for the same quality of masks, *e.g.*, 4 clicks less than InterObject3D and 2 clicks less than Inter-Object3D++ to achieve on average 80% IoU on ScanNet40 (Tab. 4, ln 1-3). The benefits of multi-object handling in AGILE3D are also validated qualitatively in Fig. 6. In this scene, after a total of 8 clicks (both methods), AGILE3D achieves an average IoU of 83.4 *vs.* 76.0 of InterObject3D++. These results highlight the benefits of interactive multi-object segmentation: (1) *Click sharing*: in AGILE3D, clicks on one object are naturally utilized to segment other objects, *e.g.*, the positive click on Chair 1 ■ (Fig. 6-g) naturally serves as a negative click for Chair 2 ■ and improves the segmentation for both objects (compare with Fig. 6-f). By contrast, in the baselines, clicks are individually given and only have an effect on that one object. Please note simply equipping the baselines with click sharing does not bring performance gains (see Appendix). (2) *Holistic reasoning*: since we segment all the objects together, AGILE3D can capture their contextual relationships, enabling holistic reasoning. For example, clicks on the armrest of one chair help correct the armrests of other chairs (Fig. 6-i,j). (3) *Globally-consistent mask*: AGILE3D encourages different regions to directly compete for space in the whole scene so that each point is assigned exactly one label. By contrast, in single-object segmentation, the mask for each object is obtained separately. Post-processing is required to generate non-overlapping masks. (4) *Faster inference*: AGILE3D pre-computes backbone features *once* per scene (∼0.05s) and runs a light-weight decoder per iteration (∼0.02s). By contrast, InterObject3D++ goes through the entire network per iteration (∼0.05s). In this scene, after 8 clicks, AGILE3D has an inference time of 0.15s *vs.* 0.4s of InterObject3D++.

**Efficiency Comparison** is shown in Tab. 5. The inference time is measured after on average 5/10/15 clicks allocated to each object in each scene. The results show that our model has comparable memory consumption but our inference time (which is much more critical for real-time applications) is *2× faster* than the baselines due to our model architecture design (Fig. 1).We report FLOPs for completeness however the results are not comparable since the MinkowskiEngine uses some clever hashing operations that can not be measured by existing FLOPs profilers.

### 4.3 USER STUDY

To go beyond simulated user clicks and assess the performance with real human clicks, we perform a user study. Tab. 6 demonstrates that real users achieve results comparable to the simulator. The results also verify that our iterative training strategy can improve the performance of both the real user and the simulator. Moreover, the variance indicates that the model trained with iterative training is more robust across different user behaviors.

In Tab. 7, we compare the results of real users in single-object and multi-object settings. We report the total given clicks per object, the achieved IoU per object, and the time spent per scene. In multi-object settings, users achieved higher IoU with significantly fewer clicks in less time, indicating that the multi-object setting can be more beneficial. Details on the user study design in the Appendix.

Table 5: **Efficiency comparison.** Our method is 2× faster than the baselines.

| Method | FLOPs/G | Mem/Mb | $t@\overline{5}$ | $t@\overline{10}$ | $t@\overline{15}$ |
|---|---|---|---|---|---|
| InterObject3D | 1.58 | 924 | 1.2 | 2.4 | 3.6 |
| InterObject3D++ | 1.58 | 924 | 1.2 | 2.4 | 3.5 |
| AGILE3D | 16.57 | 710 | **0.5** | **1.0** | **1.6** |

Table 6: **User study** on iterative vs random sampling.

| User | Training | $\overline{\text{IoU}}@\overline{3}$ ↑ | var. | $\overline{\text{NoC}}@\overline{80}$ ↓ | var. |
|---|---|---|---|---|---|
| Human | Iterative | 86.8 | 5e-5 | **1.6** | 0.02 |
| Sim. | | **88.2** | - | 2.1 | - |
| Human | Random | 85.5 | 6e-4 | 1.8 | 0.51 |
| Sim. | | 86.4 | - | 3.3 | - |

Table 7: **User study** on annotation of 30 objects.

| Settings | $\overline{\text{NoC}}$ | $\overline{\text{IoU}}$ | $\bar{t}$ |
|---|---|---|---|
| Single-object | 12.6 | 87.4 | 4min |
| Multi-object | 5.8 | 88.0 | 3min |

## 4.4 ABLATION STUDIES AND DISCUSSION

We ablate several aspects of our architecture and training in Tab. 8, 9. All the experiments are conducted on the ScanNet40 dataset on multi-object scenario unless stated otherwise. More ablations are available in the Appendix.

**Iterative training.** We validate the effectiveness of our iterative training strategy (Sec. 3.2) by training a model with randomly sampled clicks. Tab. 9, ① shows that the random sampling strategy performs noticeably worse than our iterative training (82.0 *vs.* 84.4 $\overline{\text{IoU}}@\overline{10}$). We also compare with two popular iterative strategies: ITIS (Mahadevan et al., 2018) and RITM (Sofiiuk et al., 2022). We compare in the single-object setting for a fair comparison since they are designed for such a setting. Tab. 8 shows that the model trained using our proposed training strategy significantly outperforms models trained using ITIS or RITM. More details in the Appendix.

Table 8: **Iterative training strategies** (single-object).

| Method | IoU@10 ↑ | NoC@85 ↓ |
|---|---|---|
| ITIS | 79.9 | 11.1 |
| RITM | 81.4 | 10.3 |
| AGILE3D | **82.9** | **9.8** |

**Attention design.** We design a click attention module to enable explicit interaction between the click queries themselves and between them and the point features. Tab. 9 shows that the absence of any type attention mechanism harms the model's performance. Especially, the cross-attention between click queries and point features, *i.e.*, ② C2S attn and ④ S2C attn, have a more pronounced effect compared to the self-attention between click queries ③ C2C attn. We note that C2C attn has a significant effect on the transfer experiments for KITTI-360, *e.g.* 37.4 (with C2C) *vs.* 33.8 (wo. C2C) on $\overline{\text{IoU}}@\overline{5}$, indicating that C2C plays a crucial role for improving the model's generalization ability.

Table 9: **Ablation study.**

| | Methods | $\overline{\text{IoU}}@\overline{10}$ ↑ | $\overline{\text{NoC}}@\overline{85}$ ↓ |
|---|---|---|---|
| | AGILE3D (Ours) | **84.4** | **10.7** |
| ① | − iterative training | 82.0 | 12.2 |
| ② | − C2S attn | 82.8 | 12.0 |
| ③ | − C2C attn | 84.0 | 10.8 |
| ④ | − S2C attn | 83.1 | 11.4 |
| ⑤ | − all attention | 79.2 | 13.7 |
| ⑥ | − spatial part | 84.2 | 10.8 |
| ⑦ | − temporal part | 83.9 | 10.8 |
| ⑧ | − both parts | 83.7 | 11.2 |

**Spatial-temporal encodings for click queries.** We regard user clicks as an ordered sequence of 3D coordinates and supplement each click query with a spatial-temporal encoding. Tab. 9, ⑥⑦ demonstrates that both the spatial and temporal encoding contribute to improved performance. Additionally, on KITTI-360, we note the effect of spatial-temporal encodings is even more pronounced, *e.g.* removing the temporal part led to a drop from 42.3 to 40.5 and removing the spatial part led to a drop to 36.9 on $\overline{\text{IoU}}@\overline{10}$.

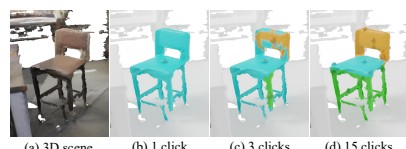

(a) 3D scene  (b) 1 click  (c) 3 clicks  (d) 15 clicks

Figure 8: **Failure cases** on fine-grained segmentation, *e.g.* part segmentation of a chair.

**Limitations and failure cases.** When users click on the seat of a chair, do they want to segment the entire chair or only the seat? Our model might not handle such ambiguity explicitly. Although AGILE3D can segment objects in its entirety very well (often with a single click as shown in Fig. 8 b), it may struggle when segmenting object parts (Fig. 8 c) and requires more clicks to segment object parts correctly (Fig. 8 d). This can be potentially solved by learning object-part relations and we leave it as future work. Second, like all interactive segmentation models, AGILE3D does not support yet a prediction of semantic labels along with the predicted mask. Equipping interactive segmentation models with semantic-awareness could be a potential future work.

## 5 CONCLUSION

We have proposed AGILE3D, the first interactive 3D segmentation model that simultaneously segments multiple objects in context. This is beyond the capabilities of existing interactive segmentation approaches, which are limited to segmenting single objects one by one. While offering faster inference, AGILE3D further achieves state-of-the-art in both interactive single- and multi-object benchmarks, in particular in the challenging low-click regime. We believe AGILE3D opens a new door for interactive multi-object 3D segmentation and can inspire further research along this line.

ACKNOWLEDGMENTS

We sincerely thank all volunteers who participated in our user study. Francis Engelmann and Theodora Kontogianni are postdoctoral research fellows at the ETH AI Center. This project is partially funded by the ETH Career Seed Award - Towards Open-World 3D Scene Understanding, NeuroSys-D (03ZU1106DA), BMBF projects 6GEM (16KISK036K) and Hasler Stiftung Grant Project (23069).

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

# Appendix

The appendix is organized as follows:

- §A: Additional model details including:
  - Architecture and detailed formulations of our click attention module.
  - Implementation details of the whole model and training procedure.
- §B: More experimental results including:
  - More qualitative results in various challenging cases.
  - Labeling new datasets without ground truth for ARKitScenes.
  - More qualitative results on KITTI-360, the outdoor LiDAR scan dataset.
  - Additional ablation study on the design choices of our query fusion strategies.
  - Additional ablation study on the term balance in our spatial-temporal query encoding.
  - Theoretical and experimental comparison of our proposed iterative training strategy with ITIS (Mahadevan et al., 2018) and RITM (Sofiiuk et al., 2022).

– Additional ablation study on extending baselines (InterObject3D and InterObject3D++) for click sharing.

– More analysis on backbone feature maps and click query attention maps.

• §C: Additional details about user studies including study designs and our developed user interface.

• §D: Details of datasets and the data preparation procedure for our interactive multi-object segmentation setup.

## A    MODEL DETAILS

### A.1    CLICK ATTENTION FORMULATIONS

The detailed architecture of our click attention module is shown in Fig. 9. All queries are represented by the positional part $Q_p$ and the content part $Q_c^l$, which are initialized through our click-as-query module. We use the same representation for each point in the 3D scene. The content part of the scene $F_c^l$ is initialized from backbone features and the positional part is obtained via Fourier positional encodings (Tancik et al., 2020). First, a click-to-scene attention module performs cross-attention from click queries to point features, which enables click queries to extract information from relevant regions in the point cloud (Eq. 1). In practice, we employ masked attention (Cheng et al., 2022), which applies an attention mask $\mathcal{H}$ to constrain the attention of each click query to the points within its intermediate predicted mask from the previous layer. Then in the click-to-click attention module, each click query self-attends to each other to realize inter-query communications (Eq. 2). To make the point features click-aware, we let the point features cross-attend to the click queries in a scene-to-click attention module (Eq. 3). In equations (1) to (3), we omit the layer normalization and dropout for simplicity. $W_Q$, $W_K$, $W_V$ are learnable weights for query, key and value as in the standard attention mechanism (Vaswani et al., 2017). In all attention modules, we add the positional part to their respective keys/queries.

$$Q_c^{l+1} = \text{softmax}\left(\frac{W_Q(Q_c^l + Q_p) \cdot W_K(F_c^l + F_p) + \mathcal{H}}{\sqrt{D}}\right) \cdot W_V F_c^l + Q_c^l \tag{1}$$

$$Q_c^{l+1} = \text{softmax}\left(\frac{W_Q(Q_c^l + Q_p) \cdot W_K(Q_c^l + Q_p)}{\sqrt{D}}\right) \cdot W_V Q_c^l + Q_c^l \tag{2}$$

$$F_c^{l+1} = \text{softmax}\left(\frac{W_Q(F_c^l + F_p) \cdot W_K(Q_c^l + Q_p)}{\sqrt{D}}\right) \cdot W_V Q_c^l + F_c^l \tag{3}$$

### A.2    IMPLEMENTATION DETAILS

**Model setting.** We use the same Minkowski Res16UNet34C (Choy et al., 2019) backbone as Kontogianni et al. (2023); Schult et al. (2023). We first quantize the 3D scene into $N'$ voxels with a fixed size of 5cm as in Kontogianni et al. (2023); Schult et al. (2023). The backbone takes as input the sparse voxelized 3D scan and produces a feature map $F \in \mathbb{R}^{N' \times 96}$. The feature maps are further projected to 128 channels by a 1×1 convolution. The click attention module consists of 3 layers with 128 channels. All attention modules in each layer have 8 heads. The feed-forward network has a feature dimension of 1024. In addition to regular background click queries generated from users' given background clicks, we also additionally use 10 learnable background queries, which are omitted in the architecture figure (Fig. 2) for visual clarity.

**Training.** We set the $\lambda_{\text{CE}} = 1$ and the $\lambda_{\text{Dice}} = 2$ in the loss function. The loss is applied to every intermediate layer of the click attention module. We use the AdamW optimizer (Loshchilov & Hutter, 2019) with a weight decay factor 1e-4. We train the model on ScanNet40 for 1100 epochs with an initial learning rate 1e-4, which is decayed by 0.1 after 1000 epochs. Due to the smaller data size, we train the model on ScanNet20 for 850 epochs with an initial learning rate 1e-4, which is decayed by 0.1 after 800 epochs. We use a single TITAN RTX GPU with 24GB memory for training.

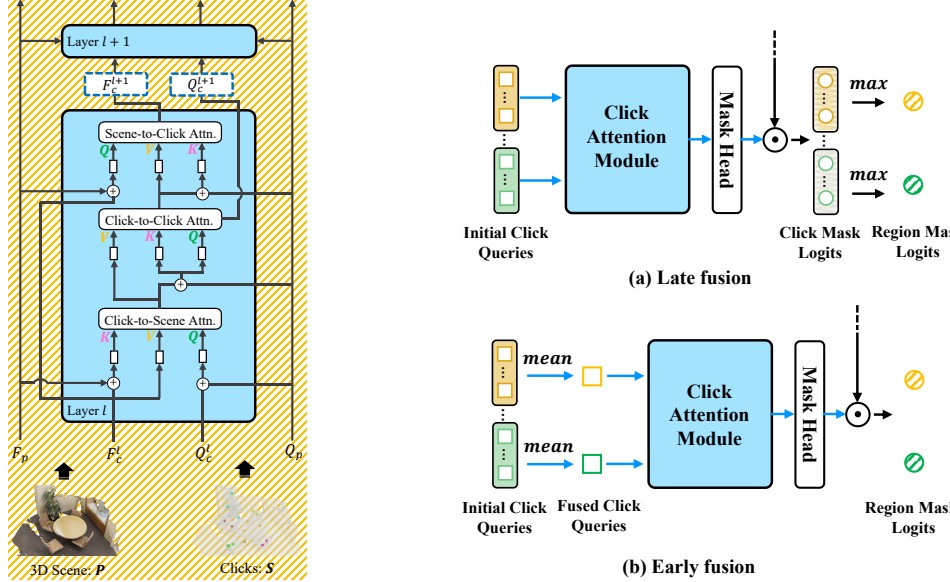

Figure 9: **Structure of the click attention module.** We omit the FFN block for clarity.

Figure 10: **Different fusion strategies.** (a) Late fusion and (b) Early fusion

## B ADDITIONAL RESULTS

### B.1 MORE QUALITATIVE RESULTS IN VARIOUS CHALLENGING CASES.

We include more qualitative results of interactive multi-object 3D segmentation in various challenging cases in Fig. 11, *e.g.*, target objects that are occluded by other objects, small objects, connected/overlapped objects, un-rigid objects, thin structures around objects, and extremely sparse outdoor scans.

### B.2 LABELING NEW DATASETS WITHOUT GROUND TRUTH FOR ARKITSCENES

In the main paper, we conducted extensive evaluations on real-world datasets ScanNet (Dai et al., 2017), S3DIS (Armeni et al., 2016) and KITTI-360 (Liao et al., 2022). To further demonstrate the practical usage of our work, we use our interactive tool to annotate a variety of challenging scenes from ARKitScenes (Baruch et al., 2021), which is a diverse real-world dataset captured with the Apple LiDAR scanner but does not contain annotations for instance masks. Here we show our generated segmentation results in Fig. 12. Although our model is only trained on ScanNet, it can generalize well to unseen datasets, which is a crucial ability to label new datasets in practice. Typically with only about 1 click on average per object (Second column), our model can already produce visually satisfying results. With more clicks, our model can further refine the segmentation and produce higher-quality masks (Third column).

### B.3 MORE QUALITATIVE RESULTS ON KITTI-360

We show more qualitative results of our method in multi-object setting on KITTI-360 in Fig. 13. KITTI-360 (Liao et al., 2022) has the largest domain gap with ScanNet40 since it is an outdoor dataset and the point cloud is much sparser. To push the limits of our model, we additionally evaluate our model on the single scan, which is even sparser. Our model is effective for multi-object segmentation on both accumulated scans and single scans. Please note we only train our model on the indoor dataset ScanNet and directly evaluate on the outdoor dataset without any fine-tuning. Those results again demonstrate the strong generalization ability of our model on datasets with large domain shifts.

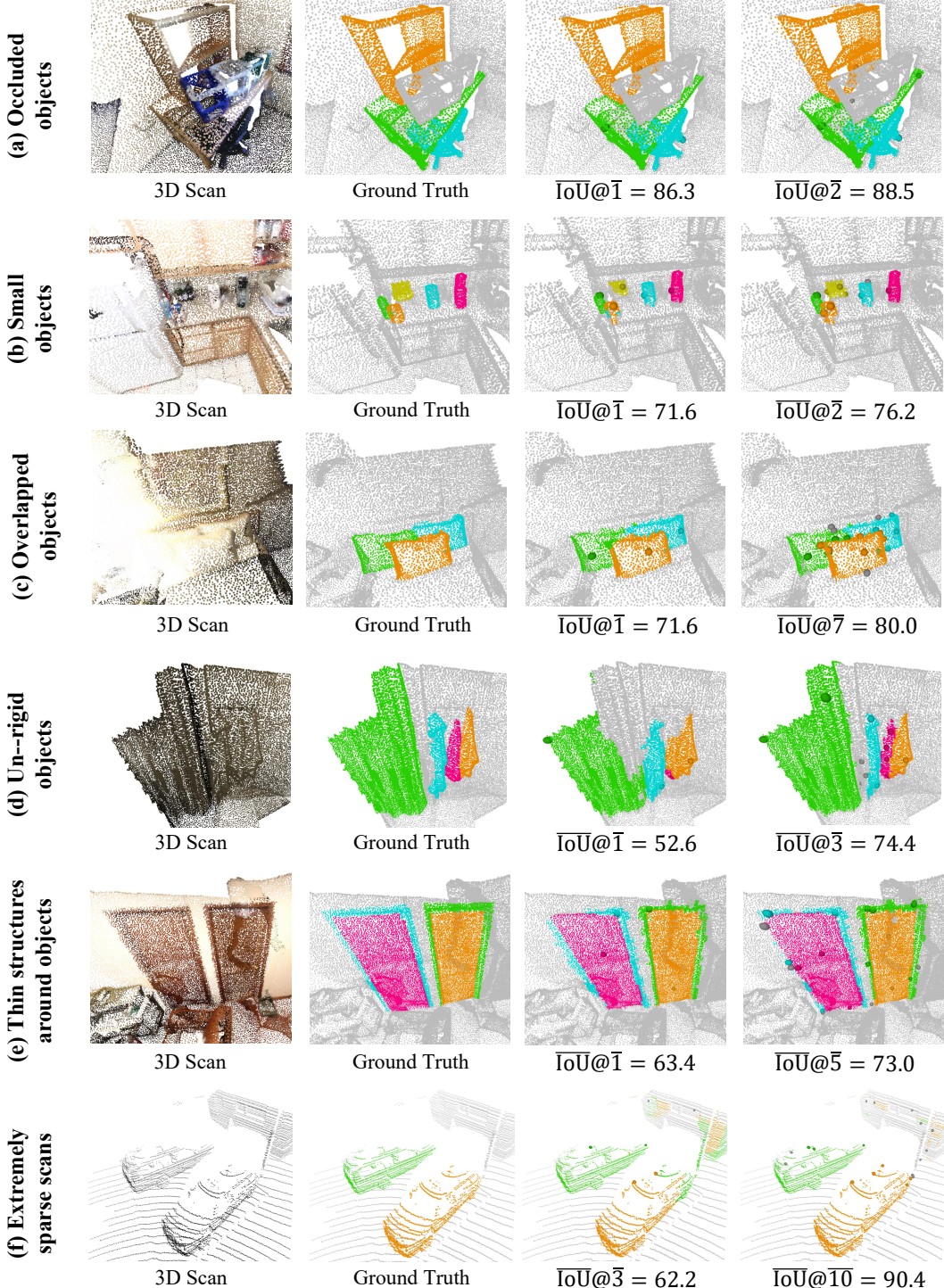

Figure 11: **More results on challenging cases in interactive multi-object segmentation.** AG-ILE3D is robust across various challenging scenarios, *e.g.*, (a) a table is largely occluded by boxes and shelf. (b) small objects like bottles and jars. (c) three pillows are connected and overlap each other. (d) un-rigid objects such as curtains and bathrobes. (e) Very thin structures like door frames around doors. (f) extremely sparse scans in outdoor scenes. Best viewed in color and zoom in.

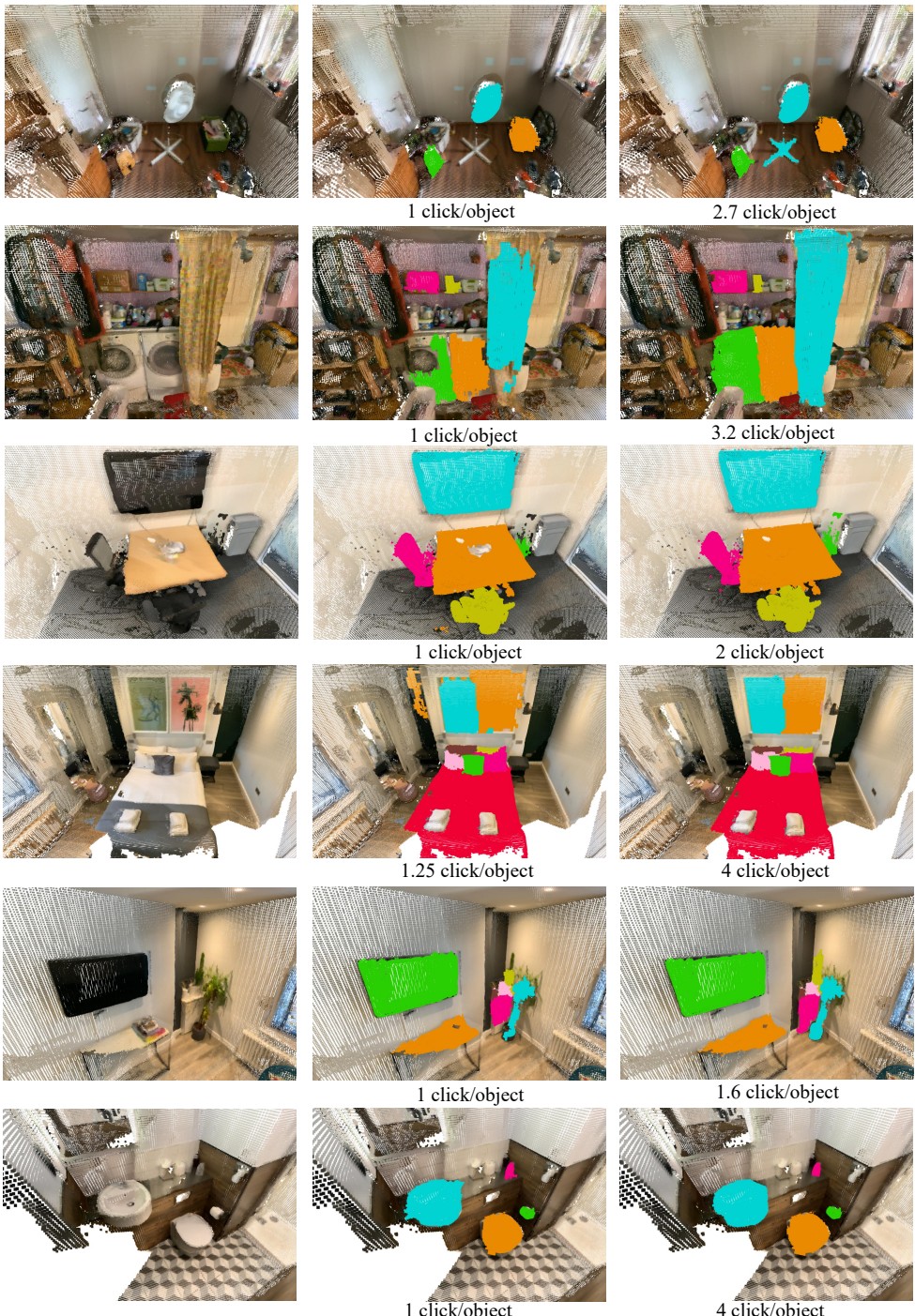

Figure 12: **Labeling new scenes for ARKitScenes** (Baruch et al., 2021). Since ARKitScenes does not contain ground truth instance masks, we show qualitative results annotated using our interactive tool. Although our model is only trained on ScanNet (Dai et al., 2017), it can be effectively used to annotate unseen datasets. Typically with only about 1 click per object, our model can already generate visually satisfying results (Second column). More clicks further improved the segmentation. The number of clicks is averaged over all target objects.

## B.4 ABLATION ON QUERY FUSION STRATEGY

In our query fusion module, we apply a per-point `max` operation to aggregate click-specific masks to region-specific masks (Fig. 10 a). To validate this design, we compare with an early fusion strategy

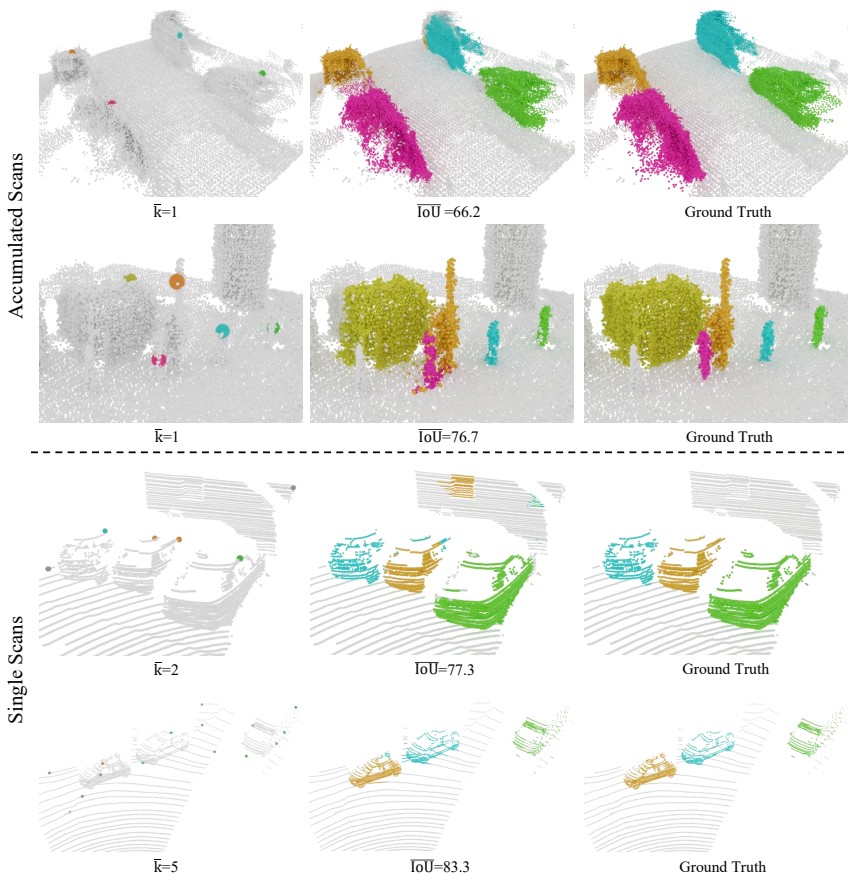

Figure 13: **Qualitative results on interactive multi-object segmentation on outdoor scans.** We evaluate AGILE3D on both accumulated scans and single scans. Even though AGILE is only trained on the indoor dataset ScanNet40, it can effectively segment outdoor scans (either accumulated or single scans).

Table 10: **Ablation on query fusion.**

| Methods | $\overline{IoU}@\overline{10}$ ↑ | $\overline{NoC}@\overline{85}$ ↓ |
|---|---|---|
| Late fusion - max (ours) | **84.4** | **10.7** |
| Late fusion - mean | 83.8 | 10.9 |
| Early fusion - max | 78.5 | 13.6 |
| Early fusion - mean | 79.0 | 13.4 |

Table 11: **Ablation on term balance in spatial-temporal query encoding.**

| Spatial:Temporal | $\overline{IoU}@\overline{10}$ ↑ | $\overline{NoC}@\overline{85}$ ↓ |
|---|---|---|
| 1:1 | 83.9 | 11.1 |
| 1:2 | 83.6 | 11.2 |
| 2:1 | 83.7 | 11.2 |
| Concatenation+Projection | 83.7 | 11.3 |

where we aggregate all initial click queries that share the same region into a single query before feeding them to our click attention module (Fig. 10 b). For both fusion strategies, we investigate both `max` and `mean` aggregation. Tab. 10 shows the late fusion strategy outperforms the early fusion strategy significantly. Although the early fusion strategy can reduce the query number and is more computationally efficient, it may cause information loss. By contrast, the late fusion strategy encourages each query to learn to represent different parts of the target object (Fig. 15). The final fusion of these queries produces a strong and global representation of the target object.

## B.5 ABLATION ON TERM BALANCE IN SPATIAL-TEMPORAL ENCODING

In our click-as-query module, we consolidate the spatial and temporal parts to a positional part by direct summation. We show the ablation on the weights of the spatial and temporal part in the summation in Tab. 11. In addition to the weighted sum, we also try to concatenate the spatial and temporal parts into a feature vector then followed by a linear projection layer. The results show our method works well with different weights. The weighted sum has similar scores with

the concatenation+projection strategy but is more efficient without introducing additional model parameters. We choose 1:1 in the main paper to avoid extensive hyperparameter tuning.

## B.6 COMPARISON OF ITERATIVE TRAINING STRATEGIES

Table 12: **Comparison of iterative training strategies.**

| Method | IoU@5 ↑ | IoU@10 ↑ | IoU@15 ↑ | NoC@80 ↓ | NoC@85 ↓ | NoC@90 ↓ |
|--------|---------|----------|----------|----------|----------|----------|
| ITIS | 74.7 | 79.9 | 81.9 | 8.7 | 11.1 | 14.2 |
| RITM | 77.7 | 81.4 | 82.9 | 7.9 | 10.3 | 13.5 |
| AGILE3D | **78.5** | **82.9** | **84.5** | **7.4** | **9.8** | **13.1** |

While acknowledging the previous exploration of iterative training in interactive 2D segmentation (Mahadevan et al., 2018; Sofiiuk et al., 2022), we underline two key distinctions in our approach: (1) Multi-object in 3D setup: Adapting the iterative training strategy directly from single-object 2D segmentation to our context is not possible. Unlike simple click sampling from false-positives or false-negatives, our strategy addresses error regions of multiple objects holistically. (2) Fully iterative sampling: Prior approaches use random sampling as initialization to avoid computational complexity and then incorporate a few iterative samples. In contrast, our training follows a fully iterative process. We start with one click per object and then sample clicks from the top error regions (one per region) in each iteration. Our method generates numerous training samples/clicks in a few iterations. Thus, it maintains reasonable training complexity while enhancing performance.

Here, we experimentally compare with previous training strategies: ITIS (Mahadevan et al., 2018) and RITM (Sofiiuk et al., 2022). The training strategy in RITM is built upon ITIS with improvements, e.g., batch-wise iterative training instead of epoch-wise iterative training. However, both methods are designed for interactive single-object image segmentation. For comparison, we adopt ITIS and RITM to train our model in single-object setting and compare with our model trained using our training strategy. The results on ScanNet40 are shown in Tab. 12.

The model trained using our proposed training strategy significantly outperforms models trained using ITIS or RITM. Please note we evaluate the model in single-object setting, which already favors ITIS and RITM since they are designed for that. The results demonstrate the superiority of our proposed iterative multi-object training scheme.

## B.7 EQUIPPING THE BASELINE WITH CLICK SHARING ABILITY

Table 13: **Additional ablations on click-sharing for the baselines on ScanNet40 dataset**

| Method | IoU@5 ↑ | IoU@10 ↑ | IoU@15 ↑ | NoC@80 ↓ | NoC@85 ↓ | NoC@90 ↓ |
|--------|---------|----------|----------|----------|----------|----------|
| InterObject3D | 75.1 | 80.3 | 81.6 | 10.2 | 13.5 | 16.6 |
| InterObject3D w. click sharing | 73.7 | 80.0 | 81.2 | 10.7 | 13.7 | 16.7 |
| InterObject3D++ | 79.2 | 82.6 | 83.3 | 8.6 | 12.4 | 15.7 |
| InterObject3D++ w. click sharing | 79.4 | 83.0 | 83.7 | 8.6 | 12.2 | 15.5 |
| **AGILE3D** (Ours) | **82.3** | **85.0** | **86.0** | **6.3** | **10.0** | **14.4** |

It is natural to ask whether we can simply equip the baseline with the click-sharing ability. We tried to adapt the baseline to be evaluated by our interactive multi-object segmentation protocol for a fair comparison in Tab. 4. Now we further modify the baseline to support click sharing: we save the positive clicks of already segmented objects and add those clicks to the negative clicks for the next object. We report the new evaluation results on ScanNet40 in Tab. 13.

The results show that this extension only brings marginal gains to InterObject3D++ and even harms the performance of the original InterObject3D. We speculate that this is due to limitations inherent in the single-object network design and training process:

(1) The baseline models encode clicks as two clicks maps: one for positive and one for negative clicks. During training, the models only see clicks related to a single object and typically those clicks are spatially close. However, if we directly equip the single-object model with click-sharing during inference, the model encounters "unrelated" clicks from other objects. This would introduce a different distribution on the clicks maps from those during training.

(2) Even though equipped with click-sharing, the baseline model still only predicts a binary mask and the masks of different objects have no direct communication. In contrast, our AGILE3D is an end-to-end model that directly predicts the multi-object masks in a holistic manner.

### B.8  MORE ANALYSIS ON BACKBONE FEATURE MAPS AND CLICK QUERY ATTENTION MAPS

**What does the backbone learn?** As we only input the 3D scene to the backbone, the backbone learns general object-level features which we visualize in Fig. 14. These features are beneficial for click queries for extracting target object features in the click-to-scene attention module.

**What does each click query attend to?** We encode clicks as queries, which undergo iterative updates by cross-attending to point features and self-attending to one another, resulting in a meaningful representation of the target object. In Fig. 15, we visualize the attention maps of each click query which reveal that each query attends to specific regions, *e.g.*, click $c_1$ attends to the entire chair with emphasis on legs, $c_2$ captures the general shape of the table while $c_3$ focuses on the nearby leg, aligning well with the user's intention to use click $c_3$ for refining the segmentation of the table leg.

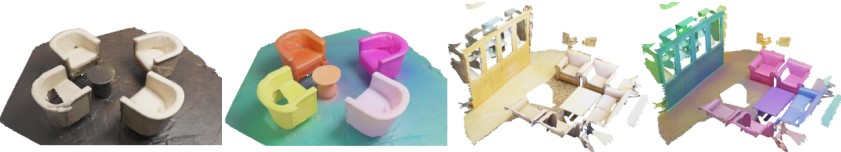

Figure 14: **Learned backbone features.** AGILE3D learns general object-level features. The point features are projected to RGB space using principal component analysis (PCA).

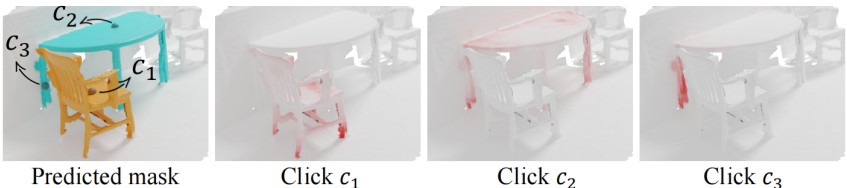

| Predicted mask | Click $c_1$ | Click $c_2$ | Click $c_3$ |

Figure 15: **Attention maps for click query $c_1$, $c_2$, $c_3$, respectively.** Each query attends to specific regions.

## C  USER STUDY

**Design.** To go beyond simulated user clicks and assess performance with clicks from real human user behavior, we perform real user studies. To that end, we first implement an interactive annotation tool with a user-friendly interface (Fig. 16). Our users were not professional labelers and they have not used such a tool before. We showed them written instructions and also explained verbally to them how the tool works. We *did not* explicitly instruct the user to click the overall max error region but instead allowed them to follow their preferences. Before recording their results, we allow the users to label an example scene to familiarize themselves with the tool. Each user is asked to annotate 5 scenes, each of which consists of 4-7 objects. Those objects cover a variety of categories, including chairs, tables, beds, lamps, telephones, etc. Please note all users labeled the same random objects for comparable results. We conducted two user studies:

(1) 12 real users are split into two groups, where 6 users use our model trained with our proposed iterative training strategy (Sec. 3.2 in the main paper), and the other 6 users use our model trained with random sampling.

(2) 12 real users are split into two groups, where 6 users annotate objects one by one in a single-object setting, and the other 6 users annotate objects in our proposed multi-object setting.

The results and analysis can be found in Sec. 4.3 in the main paper.

**User interface.** Our user interface is shown in Fig. 16 and developed based on the library Open3D (Zhou et al., 2018). The software can run across platforms and on browsers, and supports both interactive single- and multi-object segmentation. We develop various keyboard shortcuts

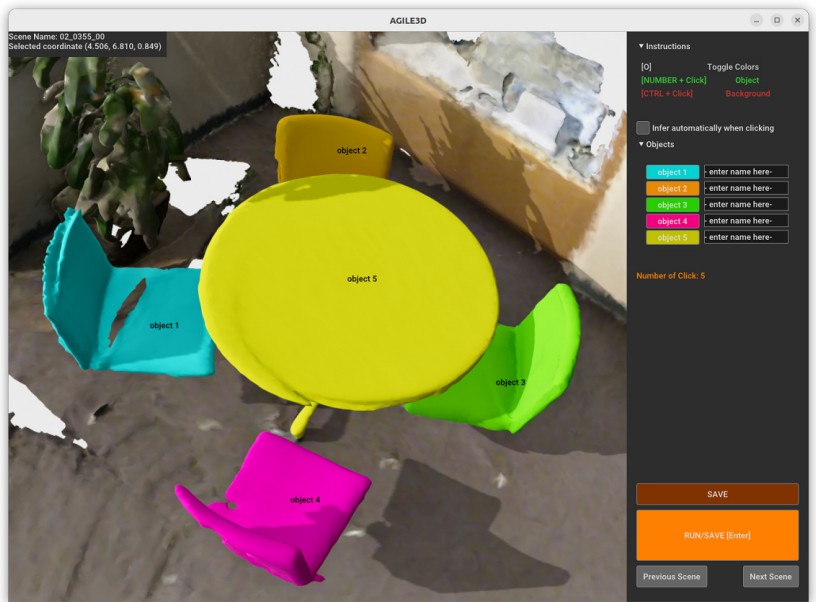

Figure 16: **User interface**

to ease user interaction, *e.g.*, Ctrl + Click would identify the background, and Number + Click would identify an object. We will release the source code of AGILE3D as well as the annotation tool to facilitate future research.

## D  BENCHMARK DETAILS

### D.1  DATASET

**ScanNetV2** (Dai et al., 2017) is a richly-annotated dataset of 3D indoor scenes, covering diverse room types such as offices, hotels, and living rooms. It contains 1202 scenes for training and 312 scenes for validation as well as 100 hidden test scenes. ScanNetV2 contains the segmentation mask for 40 classes, referred to as **ScanNet40**. However, the official benchmark evaluates only on a 20-class subset, referred to as **ScanNet20**. To test the model's generalization ability, we also use the 20-class subset training set to train the model and evaluate on both the benchmark 20 classes and the remaining unseen 20 classes.

**S3DIS** (Armeni et al., 2016) is a large-scale indoor dataset covering six areas from three campus buildings. It contains 272 scans annotated with semantic instance masks of 13 object categories. Following Kontogianni et al. (2023), we evaluate on the commonly used benchmark split ("Area 5 test").

**KITTI-360** (Liao et al., 2022) is a large-scale outdoor driving dataset with 100k laser scans in a driving distance of 73.7km. It is annotated with dense semantic and instance labels for both 3D point clouds and 2D images. The dataset consists of 11 individual sequences, each of which corresponds to a continuous driving trajectory. We evaluate the task of interactive segmentation on the sequence `2013_05_28_drive_0000_sync`.

### D.2  DATA PREPARATION

In our interactive multi-object benchmark, we aim to segment multiple objects in a scene simultaneously. However, a 3D scene may contain a large number of objects. As shown in Fig. 17, most of the scenes in ScanNetV2 (Dai et al., 2017) training set contain 20-40 objects and several scenes even contain more than 100 objects. Moreover, S3DIS (Armeni et al., 2016) contains a few very large spaces, *e.g.*, hallways with even more than millions of points. In practice, it would not be feasible for a user to simultaneously annotate all the objects in a large scene. Motivated by this considera-

tion, we set the maximum number of target objects in each evaluation sample as 10. For each scene, we randomly select $M \in [1, 2, ..., 10]$ nearby objects as evaluation samples. Since such selection involves randomness, we plan to release the object ids for a fair comparison with our method.

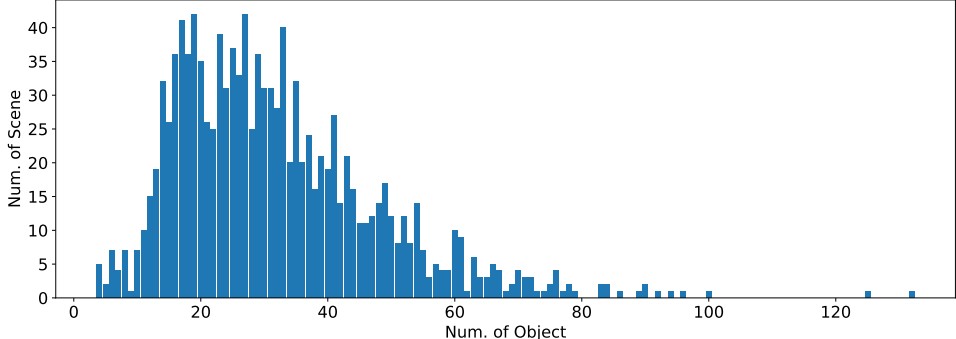

Figure 17: **Object counts in ScanNet training set**

