# OpenReview forum: "AGILE3D: Attention Guided Interactive Multi-object 3D Segmentation"
_ICLR.cc/2024/Conference — ICLR 2024 poster_

### Official Review · Reviewer_Gxfu · 2023-10-30

**Soundness:** 3 good
**Presentation:** 3 good
**Contribution:** 3 good
**Rating:** 6
**Confidence:** 3

**Summary:**

This paper introduces the novel interactive multi-object 3D segmentation task with a first multiple objects interactive segmentation approach, named AGILE3D.
The author also provides the setup, evaluation, and iterative training strategy for interactive multi-object segmentation on 3D scenes and conduct extensive experiments to validate the benefits of our task formulation.

**Strengths:**

1. The interactive multi-object 3D segmentation task is interesting and novel.
2. The author provides a complete task process, which will be helpful for subsequent research.
3. The proposed AGILE3D appears to be concise and effective, and is able to achieve state-of-the-art performance. At the same time, the author performed Efficiency Comparison, which is necessary for user-interactive tasks.

**Weaknesses:**

1. Is the contribution of this article incremental compared to 2D interactive segmentation?
Since I am not an expert in this field, I cannot judge the extent of this article's contribution.

**Questions:**

N/A

---

> ### Author Response · Authors · 2023-11-14
> **Response to Reviewer Gxfu**
>
> We thank the reviewer for the constructive feedback. We are glad to hear that the reviewer finds interactive multi-object 3D segmentation an “*interesting and novel*” task, and our model “*concise and effective*”.
> Answering the reviewer’s question, **our contribution is significant compared to 2D interactive segmentation**, which is also reflected by the comment of reviewer PRcH “*The authors offer a cogent review of pertinent literature, **spanning interactive 3D and image segmentation domains**, accentuating their contribution's novelty.*” In detail:
>
> **1. Step to multi-object handling**. Although interactive segmentation has a longer history in 2D, existing work has focussed on single-object segmentation. We show the advantage of jointly segmenting multiple objects. **The multi-object case is particularly useful in 3D, where the precise delineation of nearby and adjacent objects is harder than in an image** and we are convinced that the fresh view taken in our work will be explored further.
>
> **2. Network architecture**. Since the first deep-learning method [1] for interactive 2D segmentation, existing works have converged to a common architecture: user interactions are encoded as click maps, concatenated to the input image, and fed into a deep neural network for a binary mask prediction. Also, the state-of-the-art in 3D, InterObject3D, uses exactly that architecture (Fig. 1 in the paper). By contrast, our model encodes clicks as spatial-temporal queries and introduces a carefully designed click attention and query fusion module. This **new network layout** not only enables interactions between clicks with the scene, it also provides the flexibility needed for multi-object handling, and improves efficiency by avoiding repeated runs of the feature extraction backbone.
>
> **3. Real user study**. In 2D, the established practice is to evaluate interactive segmentation models with simulated clicks. Those simulated clicks are sampled with simple heuristics that look for the centers of the wrongly segmented regions, and do not fully reflect the actual behavior of human operators, who carefully choose where to click depending on context, contrast, point density, etc. Therefore, we have created a user interface and conducted real user studies (which support our multi-object formulation). **We hope to motivate the community in both 2D and 3D to follow that practice and go beyond the convenient, but limited proxy of simulating user clicks**.
>
> **4. Asking a new question**. We also see a contribution in preparing the ground for the 3D case. Arguably, interactive segmentation is at least as important, and more challenging in 3D, where there is a dire lack of datasets for large-scale learning, and annotation is more painful and needs automation support. In that sense, **it is also a contribution that our work formalizes the multi-object task, defines an evaluation protocol, and provides a benchmark**.
>
> [1] Xu, Ning, et al. Deep interactive object selection. CVPR. 2016.

---

> ### Author Response · Authors · 2023-11-22
> **Hope our answers are helpful**
>
> Dear reviewer Gxfu, thank you for your interest in our work. We hope you are happy with our clarifications and answers. If you have any further questions we are happy to provide further clarification.

---

### Official Review · Reviewer_3vqw · 2023-11-01

**Soundness:** 2 fair
**Presentation:** 3 good
**Contribution:** 2 fair
**Rating:** 5
**Confidence:** 4

**Summary:**

The paper proposes a new model for interactive 3D object segmentation called AGILE3D, which firstly encodes points to features. Multiple user clicks are converted to high-dimension queries, then lightweight decoder is used to segment multiple objects concurrently in 3D scene. Experimental results partly demonstrate the effectiveness of AGILE3D compared to previous models.

**Strengths:**

1. The interactive approach AGLIE3D can segment multiple objects simultaneously with limited user clicks. Compared to previous single-object iterative models, the proposed approach can reduce annotation time.
2. Sufficient experiments have been conducted to show the promising results of the proposed method.

**Weaknesses:**

1. How does the model correct wrongly segmented object instance? For example, in Figure 6, if the initial segmentation wrongly groups two chairs into one instance, how can a later click fix this mistake?
2. In Table 9, results 2 and 7 show minor performance drops without the C2C attention and temporal encoding components. Does this indicates that modeling the relations between clicks is less important for the proposed method?
3. During training and testing, the clicks are sampled at the center of the object regions. Were there any ablation experiments done using other click sampling methods? How does the choice of click sampling influence performance?

**Questions:**

Please see the weakness part.

---

> ### Author Response · Authors · 2023-11-16
> **Response to Reviewer 3vqw (Part 1/2)**
>
> We thank the reviewer for the constructive feedback. We are glad the reviewer agrees that our model is “*new*”, “*can reduce annotation time*” and is backed by “*sufficient experiments*”. Regarding the questions:
>
> **1. How does the model correct wrongly segmented object instance?**
>
> Sorry if this was unclear. If the model produces a wrongly segmented object instance, users are expected to provide assistive clicks on the error regions. Then the model will take as input the newly added clicks together with the previous clicks and the 3D point cloud and produce an updated segmentation mask. Each click $c_k = ${$p_k, o_k $}  (where $k=1, 2, …$ is the click index) has two attributes, one is its coordinates $p_k$, and another one is its region index $o_k$. $o_k = 0$ indicates the click comes from the background, $o_k=m (m>0)$ indicates the click comes from object $m$.
>
> In Figure 6, **If** the user starts with one click $c_1= ${$p_1, o_1=1$} on chair 1 but the model wrongly groups chair 1 and chair 2 into one instance, then the error region is chair 2 and the user is expected to provide a second click $c_2 = ${$p_2, o_2=2$} on chair 2. After that, $c_1$ and $c_2$ are converted to queries and go through the click attention module and each click produces one region mask logits:
>
> -$c_1$ produces $\mathcal{R}_{1} \in \mathbb{R}^{ {N}'}$, ${N}'$ is the number of points of the point cloud.
>
> -$c_2$ produces $\mathcal{R}_{2} \in \mathbb{R}^{ {N}'}$
>
> -We also have learned background mask logits $\mathcal{R}_{0} \in \mathbb{R}^{ {N}'}$.
>
> The above three mask logits are concatenated and softmaxed. Finally, an argmax will give a corrected segmentation map assigning each point a label belonging to [$0$ (background),$1$ (chair 1), $2$ (chair 2)].
>
> **2. Question on the effect of C2C attention and temporal encoding**
>
> Thanks for the reviewer’s careful observation. We agree that the self-attention between click queries (C2C attention) has a less pronounced effect compared to the cross-attention between click queries and point features, i.e., C2S attn and S2C attn. However, it does not mean that C2C attention is not important or can be safely removed. The ablation results in Table 9 are conducted in a setting where the model is trained on ScanNet40-train and evaluated on ScanNet40-val (smaller domain gap). We empirically find that C2C attn has a significant effect in the setting ScanNet40 (Training) -> KITTI-360 (Testing), **e.g. 37.4 (with C2C) vs 33.8 (wo. C2C)** on  $\overline{\textrm{IoU}}@\overline{\textrm{5}}$. Similarly, we find that the temporal encoding (TE) has an obvious effect on KITTI-360, especially for long click sequences, **e.g. 40.5 (with TE) vs 42.3 (wo. TE) on $\overline{\textrm{IoU}}@\overline{\textrm{10}}$**. This indicates that **C2C and temporal encoding play crucial roles in improving the model’s generalization ability**. **We have updated the manuscript in page 9 to highlight this point**.

---

> ### Author Response · Authors · 2023-11-16
> **Response to Reviewer 3vqw (Part 2/2)**
>
> **3. How does the choice of click sampling influence performance?**
>
> Yes, we did experiments to compare random sampling and iterative sampling.
>
> During **training**,  a naive way is **random sampling**, i.e. object clicks are randomly sampled from object regions, and background clicks are randomly sampled from the background. However, those randomly sampled clicks are **independent** of network errors, which is not consistent with testing conditions. We, therefore, propose **multi-object iterative training/sampling** to train our model, i.e. sampling new clicks from the centers of top error regions (one click per region) from the last iteration.
>
> We show the comparison of iterative training/sampling with random sampling in **Table 9** (item textcircled 1). We also compare with two iterative training strategies[1,2] proposed in the image domain in **Table 8**. We find our iterative sampling is better than both random sampling and [1,2]. Please check the **Iterative training** paragraph in **Sec. 4.4 and B.3** in the appendix for details.
>
> The established protocol for **testing** in the community is to simulate a user who always clicks at the center of the biggest error region. This protocol is deterministic and ensures reproducible scores. Like existing interactive segmentation works, we also adhere to this protocol when evaluating simulated clicks. However, we argue **this may not fully reflect real user behavior and that’s why we also conducted real user studies**. For real user studies, **we did, on purpose, not give explicit instructions on where to click, in order to obtain realistic and representative user behavior**. We conducted the user studies with two models: one trained with random sampling, one trained with our iterative training. Results in **Table 6** show that users achieve better performance with the model trained with iterative training, and exhibit smaller variance. This indicates that iterative training can indeed better mimic real user behaviors.
>
> **We hope our response have helped answer the reviewer’s questions. If so, we kindly ask the reviewer to reconsider the score. If something is still unclear, we are open to further discussions.**
>
> [1] Mahadevan, Sabarinath, et al. Iteratively trained interactive segmentation. BMVC. 2018.
>
> [2] Sofiiuk, Konstantin, et al. Reviving iterative training with mask guidance for interactive segmentation. ICIP. 2022.

---

> ### Author Response · Authors · 2023-11-22
> **A gentle reminder**
>
> Dear reviewer 3vqw, A gentle reminder, as the discussion periods will end in less than one day, we hope you are happy with our clarifications and answers. If you have any further questions, we are happy to provide further clarification. Thank you.

---

### Official Review · Reviewer_PRcH · 2023-11-01

**Soundness:** 4 excellent
**Presentation:** 4 excellent
**Contribution:** 3 good
**Rating:** 8
**Confidence:** 3

**Summary:**

The paper introduces AGILE3D, an attention-guided interactive model for multi-object 3D segmentation. At its core, AGILE3D encodes user interactions as spatial-temporal queries, leveraging a unique click attention module to foster interplay between user-generated queries and the 3D scene. The model not only supports concurrent segmentation of multiple 3D entities but also champions efficiency, demanding fewer user inputs and providing swifter inference times. Comprehensive evaluations indicate its superiority over extant methods across both individual and multi-object interactive segmentation benchmarks.

**Strengths:**

- The proposed approach pioneers in the realm of interactive multi-object 3D segmentation by introducing an avant-garde attention-centric model.
- The real-world practicality of the proposed approach is convincing with achieving top-tier results on both the popular individual and multi-object interactive segmentation benchmarks, w.r.t. SOTA methods such as Mask3D. I also appreciate the detailed computational cost comparisons.
- AGILE3D's potential to discern multiple entities with diminished user interaction and expedited inference, vis-à-vis its counterparts, is empirically substantiated.
- The authors offer a cogent review of pertinent literature, spanning interactive 3D and image segmentation domains, accentuating their contribution's novelty.
- The user study with a proper interactive interface provides convincing signals for the superiority of the model.

**Weaknesses:**

- Discussion of limitations/drawbacks should be put in the main body of the manuscript instead of the appendix for a fairer portrayal.
- More failure cases need to be showcased and analyzed if any, otherwise the paper seems to focus too much on the advantages of the presented method and does not always give the whole picture.

**Questions:**

Please check the suggestions in the Weakness section above to further strengthen the paper.

---

> ### Author Response · Authors · 2023-11-14
> **Response to Reviewer PRcH**
>
> We thank the reviewer for the constructive feedback. We appreciate the reviewer’s generally positive view, and are particularly happy that they attest our paper “novelty”, a “cogent review” and “top-tier results” with “convincing signals for the superiority”. The only real concern appears to be the discussion of limitations and failure cases.
>
> >Discussion of limitations/drawbacks should be put in the main body of the manuscript instead of the appendix for a fairer portrayal.
>
> Thank you for the suggestion. We have now moved the limitations from the appendix to the main paper.
>
> >More failure cases need to be showcased and analyzed if any, otherwise the paper seems to focus too much on the advantages of the presented method and does not always give the whole picture.
>
> Thank you, we agree that we did not pay enough attention to this. We have added failure cases and an accompanying discussion in the main paper, please see the updated manuscript for details. We hope the analysis will motivate further research into the under-explored topic of interactive 3D segmentation.

---

> ### Author Response · Authors · 2023-11-22
> **Hope our revision is helpful**
>
> Dear reviewer PRcH, thank you for your support and suggestions for our work, which further strengthen the paper. If you have any other questions, we are happy to provide more information.

---

### Official Review · Reviewer_byjT · 2023-11-01

**Soundness:** 2 fair
**Presentation:** 3 good
**Contribution:** 2 fair
**Rating:** 3
**Confidence:** 4

**Summary:**

The paper presents AGILE3D, the interactive 3D segmentation model that simultaneously segments multiple objects in context. The key idea is to use the attention-guided interactive 3D segmentation approach and also propose the click attention module to build the connection between different clicks, finally achieving multi-object segmentation via fast inference. The experimental results demonstrate and validate the proposed achievement of better performance on ScanNetV2 and outperform the competitive method InterObject3D.

**Strengths:**

1) The paper provides detailed information and resources about AGILE3D, allowing readers to gain a deeper understanding of the framework and its implementation.
2) The appendix includes additional materials such as network architecture, algorithms, and data samples, which further support the understanding and practical implementation of AGILE3D. The authors provide clear and detailed explanations of the concepts, algorithms, and techniques used in AGILE3D, ensuring that readers can follow and reproduce the framework effectively.

**Weaknesses:**

1) Lack of Novelty: For the proposed methods, the novel point is the multi-object segmentation, and the fast inference is also a novel point to some degree. And for efficiency, the submission does not present a technical point on how to improve efficiency. Multi-object segmentation is also achieved by the previous methods (InterObject3D) via some small improvements. Actually, the performance on the multi-object segmentation is not the best, according to Table 4.
2) Limited Evaluation: The appendix focuses more on providing information and supplementary materials than conducting extensive evaluations. A more thorough evaluation of the framework's performance and comparison with existing approaches would enhance the paper's credibility. The various datasets are also a strong evaluation, especially for the interactive tools. I suggest some results on the real captured and other challenging datasets should be presented in the paper.
3) The running time should be reported in a fair manner; for table 5, it is not clear how to perform the comparison since the manuscript claims efficiency as a contribution. Or, the user study is a good choice to compare the efficiency.
In some challenging cases, there are a lot of occlusions between the segmented object and other objects in the scene.
4) The significant difference between interobject3D and interobject3D++ should be discussed comprehensively since the published paper is very relative to the submission.
5) From Figure 1, the input only presents the XYZ + RGB, i.e., the point cloud and point color. Why does the visualization have the faces? Could you use the faces as extra information to enhance the performance of the proposed methods? From the interactive demo, there is still the face visualization.
6) The related works are very sparse, especially for the 3D instantce segmentations. I suggest the author refer to some surveys about the 3D segmentation and discuss more papers on it, especially for the pointnet and pointnet++.
7) For the comparison, it is not clear to me: Are the fewer user clicks the same for the different methods?
8) The failure cases and limitations are not discussed comprehensively in the submission; these parts are very important for the complete manuscripts, and some failure cases should be presented in the submission.

**Questions:**

The strengths of the paper lie in the comprehensive information provided, the inclusion of supplementary materials, and the thorough explanations. However, the lack of novelty, limited evaluation, and other weak issues Although the appendix serves its purpose as a resource for implementing AGILE3D, it does not significantly contribute to the field. Considering these strengths and weaknesses, I am negative about the submission currently, but I look forward to the response to the above questions.

see weakness

---

> ### Author Response · Authors · 2023-11-14
> **Response to Reviewer byjT [Part 1/3]**
>
> We thank the reviewer for spending time reviewing our paper, and for the advice regarding related work and failure cases. **We do, however, disagree with some of the other comments and believe they must be due to misunderstandings**. We therefore respond below to every individual question/concern:
>
> **1. Lack of Novelty**
> > For the proposed methods, the novel point is the multi-object segmentation, and the fast inference is also a novel point to some degree.
>
> We are not sure what is meant here. The reviewer starts by saying, in their first sentence, that our multi-object segmentation and fast inference are novel. But then goes on to criticize a lack of novelty. We can only point out once more that
>
> (1) as far as we know, our work is the first to do interactive multi-object segmentation of 3D point clouds.
>
> (2) our architecture is significantly different from state-of-the-art (see Fig. 1).
>
> (3) our method yields high-quality instance masks with less human effort (fewer clicks), and also offers faster inference; thus making the labeling of 3D segmentation masks much more efficient.
>
> > And for efficiency, the submission does not present a technical point on how to improve efficiency
>
> **We respectfully disagree, this is not correct.**  The efficiency of our method comes from two sources: (1) an architecture designed with fast inference in mind; and (2) more importantly, fewer user clicks due to multi-object handling. In detail:
>
> (1) our proposed **architecture** minimizes computational cost, as described in the introduction and illustrated in Fig.1: the feature backbone only needs to be run once for a point cloud, whereas clicks are processed separately, so that for every interaction one must only run the lightweight decoder (not the complete forward pass, like for instance InterObject3D). The comparison in Sec. 4.2, respectively Tab. 5 confirms that our **inference is 2× faster** than the baselines.
>
> (2) The **multi-object formulation** allows for concurrent segmentation of several objects (as opposed to prior work that processes one object at a time). Our scheme therefore needs fewer clicks than the baselines to achieve high IoU. In interactive segmentation, the crucial point is to **gain efficiency by saving user clicks**.
>
> In this context **we note that our view is stand by all other reviewers**:
>
> -reviewer **PRcH**: *“The model ... champions efficiency, demanding fewer user inputs and providing swifter inference times”, “detailed computational cost comparisons”, “diminished user interaction and expedited inference ... is empirically substantiated.”*
>
> -reviewer **3vqw**: *“the proposed approach can reduce annotation time”*
>
> -reviewer **Gxfu**: *“the author performed Efficiency Comparison, which is necessary for user-interactive tasks.”*
>
> >Multi-object segmentation is also achieved by the previous methods (InterObject3D) via some small improvements.
>
> **Also here we disagree**. Perhaps there was misunderstanding because we adapted InterObject3D so that it could serve as a baseline? Since there isn’t any prior work about multi-object interactive segmentation, we had no baseline to compare to. Therefore, we upgraded InterObject3D, the state-of-the-art for single-object segmentation, such that it could be evaluated by our protocol. The “baseline” paragraph in Sec. 4 explains our modifications. But although we modified InterObject3D for evaluation on our multi-object benchmark **it can, by design, only predict a binary mask for a single object** in every pass through the network, and those masks must be merged into a multi-object segmentation either heuristically or by hand.
>
> >Actually, the performance on the multi-object segmentation is not the best, according to Table 4.
>
> **This is not true**. We are not sure how the reviewer arrived at that conclusion. The fact is that **AGILE3D outperforms previous methods** by a significant margin reaching “*top-tier results*” [reviewer **PRcH**] and “*state-of-the-art performance*” [reviewer **Gxfu**]. Table 4 shows 6 metrics for each of the 3 datasets. Among the 18 comparisons in total, AGILE3D outperforms standard InterObject3D **in all cases**, and outperforms our enhanced InterObject3D++ in **17 out of 18 cases**. Please do not only check the metrics in isolation but overall for KITTI-360, in that particular case AGILE3D surpasses the baseline by **a factor of 4** and the enhanced baseline by **a factor of 2** on $\overline{\textrm{IoU}}@\overline{\textrm{5}}$.

---

> > ### Comment · Reviewer_byjT · 2023-11-15
> > **response**
> >
> > In Figure 1, the illustration also claims that InterObject3D also supports interactions, which conflicts with the second contribution "the first interactive approach".
> > And I agree with the fast inference of the proposed method from the reported numbers, but I don't see any technical points that are introduced in the paper to support the fast inference.
> > For InterObject3D, the methods can be used to finish the tasks, but the only different point of the proposed methods is the fast inference. And the paper is not focused on efficiency, and this part is only small.
> > For the performance in table 4, the reported number is not very significant over the previous methods (ie. just 0.x for NoC). And also, all the reported numbers are not the best, such as IoU@15 for KITTI-360 test data.

---

> > > ### Author Response · Authors · 2023-11-17
> > > **Further clarification on interactive multi-object segmentation (Part 1/2)**
> > >
> > > > In Figure 1, the illustration also claims that InterObject3D also supports interactions, which conflicts with the second contribution "the first interactive approach".
> > >
> > > **This is not correct.** We hope the reviewer can check Figure 1 carefully and be precise. Our stated contribution is "the first interactive approach that can segment **multiple objects** in a 3D scene", "the first approach that supports interactive **multi-object** segmentation in 3D point clouds". **We never state anywhere "the first interactive approach" solely in the absence of the context of multi-object.** InterObject3D is the first interactive approach for interactive **single-object** 3D segmentation. **Figure 1** clearly shows InterObject3D can only support a **binary mask prediction for single-object** segmentation but **our model directly predicts multi-object mask**.
> > >
> > > ### **We note our contribution regarding interactive multi-object 3D segmentation is consistently recognized by other reviewers**:
> > >  **PRcH**: “*The proposed approach pioneers in the realm of **interactive multi-object 3D segmentation***”; **3vqw**: “*... AGLIE3D can segment **multiple objects simultaneously** with limited user clicks. Compared to **previous single-object iterative models**, the proposed approach can reduce annotation time.*”; **Gxfu**: “This paper introduces the **novel interactive multi-object 3D segmentation task with a first multiple objects interactive segmentation approach**, named AGILE3D.”
> > >
> > > > I don't see any technical points that are introduced in the paper to support the fast inference.
> > >
> > > In our paper and “Response to Reviewer byjT [Part 1/3]”, we explicitly explain **the technical points for fast inference comes from the architecture design**. We reiterate: we only feed point cloud to the backbone and process clicks in a lightweight decoder. This enables us to **run the backbone (0.05s) only once and then only need to rerun a lightweight decoder (0.02s) for each iteration**. By contrast, **InterObject3D needs to rerun a full forward pass of the entire network (0.05s) for each iteration**. In addition, we use queries (i.e. sparse vectors) to represent clicks, other than dense positive/negative click masks as in the baseline, which helps us design such architecture for fast inference. As a result, in Fig. 1, for 10 clicks, the inference of InterObject3D needs 0.05s × 10 = **0.5s**. The inference of our method needs 0.05s × 1 + 0.02s ×10 = **0.25s**.
> > >
> > > ### **Could the reviewer let us know why you think our above explanation is not the technical point for fast inference?**
> > > In addition, now we have used **orange color** to mark all the relevant explanations on technical points for fast inference in the paper, which hopefully can help the reviewer notice them more easily.
> > >
> > > > For InterObject3D, the methods can be used to finish the tasks, but the only different point of the proposed methods is the fast inference.
> > >
> > > **This is not correct.** First, InterObject3D and our enhanced baseline (InterObject3D++) can **only perform interactive single-object segmentation**. By design, their networks can only predict a **binary mask**. To make them comparable with us, we can only **run them for each single object separately** and finally combine all the binary masks to a multi-object mask by hand. **It is wrong to regard the above hand-crafted process as interactive multi-object segmentation** since it is essentially different from our task formulation and our proposed end-to-end method that directly predicts the multi-object masks. We have revised the paper to make this point clearer.
> > >
> > > **Second, we disagree with the unjustified point** “the only different point of the proposed methods is the fast inference. ” **which ignores our better achieved IoU and fewer NoC as well as other advantages**. We explicitly explain the **advantages of our interactive multi-object segmentation** over the hand-crafted process of the baselines in **sec. 4.2** from **four aspects**: (1) **Click sharing** (2) **Holistic reasoning** (3) **Globally-consistent mask** (4) **Faster inference**. We have highlighted that part in **orange color** and hope that can alleviate the reviewer’s misunderstanding.
> > >
> > > ### **If the reviewer believes the baselines can predict multi-object mask end-to-end like us and also has the above (1)(2)(3)(4) features, could you present any justified evidence?**

---

> ### Author Response · Authors · 2023-11-14
> **Response to Reviewer byjT [Part 2/3]**
>
> **2. Limited Evaluation**
> > The appendix focuses more on providing information and supplementary materials than conducting extensive evaluations. A more thorough evaluation of the framework's performance and comparison with existing approaches would enhance the paper's credibility. The various datasets are also a strong evaluation, especially for the interactive tools. I suggest some results on the real captured and other challenging datasets should be presented in the paper.
>
> We do not see why our evaluation would be lacking, and **our sufficient evaluation is appreciated by all other reviewers**: **PRcH**: “*Comprehensive evaluations indicate its superiority …*”; **3vqw**: “*Sufficient experiments have been conducted …*”; **Gxfu**: “*The author … conduct extensive experiments to validate the benefits …*”.
>
> We evaluate not only existing (single-object) interactive segmentation benchmark, and also provide a new multi-object benchmark. Both benchmarks include **multiple datasets of real scans**, among them **both indoor (ScanNet, S3DIS) and outdoor scans (KITTI-360)** that are all commonly used to evaluate 3D scene understanding. Since interactive 3D segmentation has only been studied recently, we have also run a state-of-the-art fully-supervised method (Mask3D) for a **complete and credible evaluation**. Finally, we have conducted **real user studies** to rule out biases due to simulated clicks, where real users worked with our interactive tools to annotate 5 real scenes that contain a variety of object categories. We speculate that the reviewer might be missing more qualitative results and have therefore moved them from the appendix to the main paper.
>
>
> **3. The running time should be reported in a fair manner**
>
> We fully agree, and did our best to ensure a fair comparison. Run times for all methods were measured in the **same setting** on the **same hardware** (a single TITAN RTX GPU).
>
> >for table 5, it is not clear how to perform the comparison since the manuscript claims efficiency as a contribution.
>
> The paragraph “efficiency comparison” in Sec. 4.2 describes this. **Inference time is measured after all objects in a given scene have received 5/10/15 clicks on average**. E.g., for a scene with 6 objects the t@$\overline{5}$ is the total time spent on forward passes through the network after 6×5=30 clicks. Throughout, our method is **2× times faster** (1.0s vs 2.4s at t@$\overline{10}$; 1.6s vs 3.5s at t@$\overline{15}$; etc.).
>
> > Or, the user study is a good choice to compare the efficiency.
>
> We fully agree, that is why we have conducted a real user study, see Tab. 7. Each user was asked to annotate 5 scenes with, in total, 30 objects. We report the final number of clicks ($\overline{\textrm{NoC}}$, average of all objects), the intersection-over-union achieved with those clicks ($\overline{\textrm{IoU}}$, average of all objects), and **the time spent by the user** to obtain that result ($\overline{\textrm{t}}$, average of all scenes). In the study, users achieved higher IoU with significantly fewer clicks, and in shorter times, with our proposed multi-object segmentation approach. I.e., the **user study confirms better efficiency with AGILE3D**.
>
> >In some challenging cases, there are a lot of occlusions between the segmented object and other objects in the scene.
>
> We are not sure what the question is. Regarding missing data due to sensor occlusions: yes, the data has the characteristics of real scans. Regarding occlusions in the visualization: with our interactive tool users can use the mouse to rotate/translate/zoom in/zoom as needed to minimize occlusions. **All users quickly got familiar with the tool and we did not receive any feedback about occlusion problems**.

---

> > ### Comment · Reviewer_byjT · 2023-11-15
> > **response**
> >
> > I just curious about more challenging cases, such as when the segmented object is occluded by other objects, and more real-world captured data. Over all the presented  results, I don't see the diversity of the dataset that is evaluated by the proposed methods; there are only few examples.

---

> ### Author Response · Authors · 2023-11-14
> **Response to Reviewer byjT [Part 3/3]**
>
> **4. The difference between interobject3D and interobject3D++**
>
> The difference is explained in the “baselines” paragraph in Sec. 4. **InterObject3D is the only recent published work** about (single-object) interactive 3D segmentation. **InterObject3D++ is our enhanced version** (i.e., a harder baseline) that we have derived from it. The reason for introducing it is that the iterative training strategy that we propose in our paper can be readily applied also to InterObject3D. **We found that, indeed, iterative training significantly improved InterObject3D. We call the enhanced version InterObject3D++**, and include it for a complete and fair comparison that properly disentangles our model design from the training strategy. **AGILE3D beats also InterObject3D++, indicating that its advantage is due to the multi-object model design and not only the result of a better engineered training strategy**.
>
> **5. Visualizations have the faces**
>
> For all visualizations of ScanNet, e.g., Fig. 1 and for the interactive demo (Fig. 12), we visualize the meshes. This is common practice, and only serves to make the renderings easier to read. E.g., [1,2,3] also plot faces, but do not use them in their computational methods. We also do not use face information since not every 3D dataset contains faces. It is of course true that one could potentially calculate normals from the faces and use them as additional input to the network, but that would mean that the network could no longer be applied to scan data that does not contain faces (like S3DIS, KITTI-360, etc.).
>
> [1] Peng, Songyou, et al. OpenScene: 3D Scene Understanding with Open Vocabularies. CVPR. 2023.
>
> [2] Schult, Jonas, et al. Mask3D for 3D Semantic Instance Segmentation. ICRA. 2023.
>
> [3] Chibane, Julian, et al. Box2Mask: Weakly Supervised 3D Semantic Instance Segmentation using Bounding Boxes. ECCV, 2022.
>
> **6. Related works**
>
> Thank you for the suggestion. We have rewritten the related work part for 3D instance segmentation, including more discussions on point-based and voxel-based learning backbones and more details on existing methods on 3D instance segmentations, as well as references to surveys. Please check the red-marked content in our revised manuscript.
>
> **7. Are the fewer user clicks the same for the different methods?**
>
> For comparison using simulated clicks, only the first click is the same for all methods. This serves as a fair, common starting point. The following clicks will be sampled from regions where a method makes mistakes, which obviously differ between methods.
>
> In the study with real user clicks, even the first click could be different, as it would defeat the purpose of a realistic user study to enforce specific click locations for everyone. We did, on purpose, also not give explicit instructions how to click, in order to obtain realistic and representative user behavior.
>
> **8. Advice on failure cases and limitations**
>
> We thank the reviewer for this suggestion. We previously had put the discussion of limitations in the appendix, but have now moved it to the main paper. Moreover, we have also added discussions on failure cases, which can hopefully motivate future work. Please check the red-marked content in our revised manuscript. We believe this update has improved the manuscript.
>
> ### **To conclude, we regret if the rather under-explored topic of interactive 3D object segmentation, and the associated lack of a canonical “standard vocabulary”, has caused misunderstandings. We hope our explanations will alleviate the reviewer’s concerns - if yes please reconsider the overall score. If there are further questions and concerns, we are open to continuing the discussion.**

---

> ### Author Response · Authors · 2023-11-17
> **Further clarification on interactive multi-object segmentation (Part 2/2)**
>
> > For the performance in table 4, the reported number is not very significant over the previous methods (ie. just 0.x for NoC). And also, all the reported numbers are not the best, such as IoU@15 for KITTI-360 test data.
>
> Here we show the results on table 4 and **explicitly mark the improvement** of our method over baselines in brackets.
>
> Table 4. Quantitative results on interactive multi-object segmentation. We adapt the state-of-the-art method in interactive single-object segmentation to be evaluated by our multi-object protocol for a complete comparison (Baseline paragraph of Sec. 4). **Note the baselines still predict binary masks for single-object and final masks must be merged manually**.
>
> | Methods |  Train $\rightarrow$ Eval |   $\overline{\textrm{IoU}}@\overline{\textrm{5}} \uparrow$ |  $\overline{\textrm{IoU}}@\overline{\textrm{10}} \uparrow$| $\overline{\textrm{IoU}}@\overline{\textrm{15}} \uparrow$|$\overline{\textrm{NoC}}@\overline{\textrm{80}} \downarrow$ |  $\overline{\textrm{NoC}}@\overline{\textrm{85}} \downarrow$| $\overline{\textrm{NoC}}@\overline{\textrm{90}} \downarrow$|
> |----------|:------:|:------:|:------:|:------:|:------:|:------:|:------:|
> | InterObject3D |  | 75.1 | 80.3 | 81.6 | 10.2 | 13.5 | 16.6|
> | InterObject3D++ | ScanNet40 $\rightarrow$ ScanNet40 |79.2 | 82.6 | 83.3 | 8.6 | 12.4 | 15.7 |
> | **AGILE3D (ours)** |   |**82.3 (+3.1)** | **85.0 (+2.4)** | **86.0 (+2.7)** | **6.3 (-2.3)** | **10.0 (-2.4)** | **14.4 (-1.3)** |
> | InterObject3D |  | 76.9 | 85.0| 87.3 | 6.8 | 8.8 | 13.5|
> | InterObject3D++ |  ScanNet40 $\rightarrow$ S3DIS-A5 |81.9| 88.3 | 89.3 | 5.7 | 7.6 | 11.6 |
> | **AGILE3D (ours)** |   |**86.3 (+4.4)** | 88.3 | **90.3 (+1.0)** | **3.4 (-2.3)** | **5.7 (-1.9)** | **9.6 (-2.0)** |
> | InterObject3D |  | 10.5 | 22.1| 31.0 | 19.8 | 19.8 | 19.9|
> | InterObject3D++ |  ScanNet40 $\rightarrow$  KITTI-360 |16.7| 37.1 | 52.2 | 18.3 | 18.9 | 19.3 |
> | **AGILE3D (ours)** | |**40.5 (+23.8)** | **44.3 (+7.2)** | 48.2 (-4.0) | **17.4 (-0.9)** | **18.3 (-0.6)** | **18.8 (-0.5)** |
>
> First, we agree the gap of NoC between our method and baselines on  KITTI-360 is not as significant as on ScanNet40 and S3DIS. However, **it is not correct to assume our method is not significant over the baselines**. We follow standard protocol in the interactive segmentation community to **threshold NoC metrics at 20 (to prevent unbounded click budgets)**, i.e. if one method didn’t achieve 80 IoU after 20 clicks, then NoC@80 was thresholded at 20. Due to the biggest domain gap lying in KITTI-360, all the methods can have cases that didn’t achieve the required IoU after 20 clicks. Even if we can achieve 80 IoU with 30 clicks but another one with 40, both of them are thresholded to 20 clicks, which would obviously narrow down their actual gaps. **To give the overall picture, here we also show NoC@50, 65, 70 and the challenging low-click regime IoU@1, 2, 3, which further verifies that we outperform baselines by a large margin.**
>
> Table 14. More results on low IoU regime and low lick regime on KITTI-360, which supplements tab. 4 to give an overall picture.
> | Methods |  Train $\rightarrow$ Eval |   $\overline{\textrm{IoU}}@\overline{\textrm{1}} \uparrow$ |  $\overline{\textrm{IoU}}@\overline{\textrm{2}} \uparrow$| $\overline{\textrm{IoU}}@\overline{\textrm{3}} \uparrow$|$\overline{\textrm{NoC}}@\overline{\textrm{50}} \downarrow$ |  $\overline{\textrm{NoC}}@\overline{\textrm{60}} \downarrow$| $\overline{\textrm{NoC}}@\overline{\textrm{70}} \downarrow$|
> |----------|:------:|:------:|:------:|:------:|:------:|:------:|:------:|
> | InterObject3D |  | 1.9 | 3.7 | 6.0 | 17.8 | 18.8 | 19.3|
> | InterObject3D++ |  ScanNet40 $\rightarrow$  KITTI-360 |3.7| 5.8 | 8.8 | 13.8 | 15.4 | 16.9 |
> | **AGILE3D (ours)** | |**34.9 (+31.2)** | **38.1 (+32.3)** | **39.4 (+30.6)** | **10.4 (-3.4)** | **13.1 (-2.3)** | **14.3 (-2.6)** |
>
> ### **We kindly ask the reviewer to consider all the results as a whole - if the reviewer thinks our method is not the best, then which one is?**

---

> > ### Comment · Reviewer_byjT · 2023-11-17
> > **response**
> >
> > I still have some concern about the overclaim for performance over InterObject3D++. From all reported numbers, the performance cannot achieve the best one fully (mentioned before), and for the evaluations, the presented visual result cannot cover the diversity of datasets. Hence, the manuscript are not above the acceptance bar of the conference and I will keep my origianl score. I am very happy to listen to the opinions of other reviewers.
> > I suggest adding more data to evaluate the proposed methods during revision.

---

> > > ### Author Response · Authors · 2023-11-20
> > > **Further clarification on performance and evaluation**
> > >
> > > We are relieved to see the reviewer’s concerns narrowed down to two concrete points: (1) our method didn’t beat InterObject3D++ on all metrics according to Tab. 4. (2) the presented visual results didn’t cover the diversity of datasets.
> > >
> > > For (1), our method outperforms InterObject3D (published baseline) on **all metrics on all three datasets**. As a matter of good scientific practice, we have ourselves constructed the enhanced baseline InterObject3D++, and our method also outperforms InterObject3D++ on **all metrics** on ScanNet and S3DIS and **only lags behind it in one single metric** on one dataset, KITTI-360. In our view this still supports our work, in line with other reviewers (“top-tier results” [PRcH], “state-of-the-art performance” [Gxfu]). We also note that, in our view and according to the ICLR review guideline (https://iclr.cc/Conferences/2024/ReviewerGuide), **performance numbers alone are not grounds for rejection**. Even without looking at the “bold numbers”, we believe that our work has several attractive properties, including **(1) faster inference, (2) click sharing, (3) globally consistent masks, and  (4) holistic reasoning (see sec. 4.2)**. In our humble view, systematically exploring interactive multi-object 3D segmentation, and introducing the first end-to-end approach for it, would be interesting contributions in addition to our overall better numbers. We are also hopeful that our new look on interactive 3D segmentation may inspire further work about interaction with 3D multi-object scenes.
> > >
> > > For (2), in the last revision, we already added **an additional page of visual results (Fig. 11 in page 15 in the appendix), with various challenging cases**: target objects that are occluded by other objects, small objects, overlapped objects, un-rigid objects, thin structures around objects, and extremely sparse outdoor scans. **We have now also included more visual results in the main paper (Fig. 5 and Fig. 7).** **Our visual results now cover various scenes from ScanNet, S3DIS and KITTI-360.** We kindly ask the reviewer to check them in our latest manuscript. If the reviewer still feels that they do not adequately cover the diversity of datasets, please let us know what other cases to include.

---

> > > ### Author Response · Authors · 2023-11-20
> > > **Updates on evaluation**
> > >
> > > Dear reviewer,
> > >
> > > In addition to our added more visual results on ScanNet, S3DIS and KITTI-360, here we additionally use our interactive tool to annotate a variety of challenging scenes from  **ARKitScenes** [1]: a dataset captured by iPad/iPhone LiDAR scanner. Unfortunately, ARKitScenes does not contain ground truth segmentation masks and cannot be used by the 3D segmentation community for evaluation. Instead, we use this dataset to further demonstrate the practicality of our model in annotating new datasets even though we didn't train on them. We have included **one more page of visual results** on our generated segmentation masks for ARKitScenes with the interactive tool in Fig. 12, page 17. Typically with one click per object, our model can already offer visually satisfying segmentation masks.
> > >
> > >
> > > Now we summarize our revisions regarding evaluations:
> > >
> > > 1. More visual results of single-object segmentation on S3DIS and KITTI-360. (**Fig. 5,  page 7**)
> > > 2. More visual results of multi-object segmentation on S3DIS and KITTI-360. (**Fig. 7, page 8**)
> > > 3. One more page of visual results of various challenging cases on both indoor and outdoor scans. (**Fig. 11, page 15**)
> > > 4. One more page of visual results of labeling new datasets for ARKitScenes. (**Fig. 12, page 17**)
> > > 5. We also suggest checking visual results on both accumulated and single scans of outdoor scenes. (**Fig. 13, page 18**)
> > >
> > > We kindly ask the reviewer to check them in our latest manuscript. If the reviewer still has concerns, we are open to further discussion.
> > >
> > >
> > > [1] Baruch, Gilad, et al. ARKitScenes - A Diverse Real-World Dataset for 3D Indoor Scene Understanding Using Mobile RGB-D Data. NeurIPS Datasets and Benchmarks Track (2021).

---

> ### Author Response · Authors · 2023-11-17
> **Further clarification on evaluation**
>
> > I just curious about more challenging cases, such as when the segmented object is occluded by other objects, and more real-world captured data. Over all the presented results, I don't see the diversity of the dataset that is evaluated by the proposed methods; there are only few examples.
>
> First, the fact is that our evaluation datasets **ScanNet, S3DIS, and KITTI-360 are all challenging real-world captured datasets, widely benchmarked in literature.**
>
> Second, **we want to know the justified reasons from the reviewer why you think ScanNet, S3DIS, and KITTI-360 are not challenging and not real-world enough**. Is there another dataset that you believe will offer something more significant than these real-world datasets we evaluated already? If yes please let us know which one and why it is better than the ones we used already.
>
> ### **Now we included one-page more qualitative results (see Fig. 11 in page 15) in various challenging cases.**
>
> We further included **one page more qualitative results in various challenging cases: target objects that are occluded by other objects, small objects, connected/overlapped objects, un-rigid objects, thin structures around objects, and extremely sparse outdoor scans**. Note here we visualize directly on the raw point clouds (not mesh). We hope now the reviewer can realize the challenging nature of the datasets we used. Now we included them in the appendix due to the limited pages in the main paper. If the reviewer recognizes our added results, we will add them to the main paper by squeezing more space.
>
> ### **To conclude, regarding the reviewer’s two main concerns on interactive multi-object 3D segmentation and evaluation:**
> (1) The **baseline** method can only predict **binary masks for single objects** by its network design and final masks must be merged by hand. **It can not be seen as interactive multi-object 3D segmentation**. **Our method is the first approach that supports interactive multi-object 3D segmentation, directly predicting multi-object masks end-to-end, offering high-efficiency, SOTA results, which are also recognized by other reviewers.**
>
> (2) **Our extensive evaluations on challenging and real-world captured datasetsScanNet, S3DIS, and KITTI-360 are recognized by all other reviewers**. **Now we further revised the manuscript to include one-page more qualitative results in various challenging cases.** Also if the reviewer has in mind other real-world datasets that can offer more insight than the ones we used already please let us know and we will add it in the final version of the paper.
>
> **If the reviewer still has concerns, we are open to further discussions.**

---

### Author Response · Authors · 2023-11-21
**Common response to all reviewers and area chairs**

Dear Reviewers and Area Chairs,

We express our gratitude to all of you for investing time in assessing our paper. We appreciate all the reviewers for their constructive comments and valuable feedback.

 We are glad that the reviewers recognize our work:
- **Task**: “*The interactive multi-object 3D segmentation task is interesting and novel*” [Gxfu], “*The author provides a complete task process, which will be helpful for subsequent research*” [Gxfu], “*pioneers in the realm of interactive multi-object 3D segmentation*” [PRcH], “*the novel point is the multi-object segmentation*” [byjT]
- **Method**: “*an avant-garde attention-centric model*” [PRcH], “*a new model for interactive 3D object segmentation*” [3vqw], “*a first multiple objects interactive segmentation approach*” [Gxfu], “*concise and effective*” [Gxfu]
- **Evaluation**: “*Comprehensive evaluations indicate its superiority*” [PRcH], “*Sufficient experiments have been conducted …*” [3vqw], “*The author … conduct extensive experiments to validate the benefits …*” [Gxfu], “*The user study … provides convincing signals for the superiority …*” [PRcH],  “*detailed computational cost comparisons*” [PRcH], “*diminished user interaction and expedited inference ... is empirically substantiated.*” [PRcH]
- **Performance**: “*top-tier results on both the popular individual and multi-object interactive segmentation benchmarks*” [PRcH], “*state-of-the-art performance*” [Gxfu], “*champions efficiency, demanding fewer user inputs and providing swifter inference times*” [PRcH], “*Compared to previous single-object iterative models, the proposed approach can reduce annotation time*” [3vqw], “*fast inference*” [byjT]

Regarding the reviewers’ concerns/questions, we have responded to them individually and made the following revisions to the manuscript. We use **red** color to indicate newly added content:


1. We added a discussion on the limitations and failure cases in the main paper (Sec. 4.4) [byjT, PRcH]
2. We added more visual results to cover the diversity of the datasets and enhance the evaluation [byjT]:
 - More visual results of single-object segmentation on S3DIS and KITTI-360. (Fig. 5, page 7)
 - More visual results of multi-object segmentation on S3DIS and KITTI-360. (Fig. 7, page 8)
 - One more page of visual results of various challenging cases on both indoor and outdoor scans. (Fig. 11, page 15)
 - One more page of visual results of labeling new datasets for ARKitScenes. (Fig. 12, page 17)
3. We revised the related work section to add more works in 3D segmentation [byjT]
4. We added the effect of C2C attention and temporal encoding on KITTI-360 to demonstrate they play crucial roles in improving the model’s generalization ability (Sec. 4.4) [3vqw]
5. We added more details to make it clearer that although we evaluated single-object baselines for a more complete comparison with our method (Baseline paragraph above sec. 4.1), their binary masks need to be merged manually (see Fig. 6, 7). This is to address the misunderstanding that assumes the baselines can also perform interactive multi-object segmentation [byjT]
6. We also use **yellow** color to highlight content on our technical point for fast inference and advantages over single-object baselines [byjT]. We will change the color-marked content back to black color in the final version of the paper.

In addition, we will make our **code** and our developed **interactive 3D annotation tool** public once the paper is accepted. We hope our work will produce practical benefits for the 3D annotation community.

We thank again the reviewers for the constructive feedback, which helped improve the quality of the paper. If the reviewers find our rebuttal helpful, we sincerely hope they consider raising their scores. If the reviewers have any further concerns or questions, please let us know before the discussion period ends.

Best regards,

Paper 5206 authors

---

### Meta-Review · Area_Chair_bpFd · 2023-12-09

**Metareview:**

This paper addresses interactive segmentation of 3D point clouds and is the first framework to support multiple objects.  The reviews are diverse, spread between reject, borderline reject & accept, and accept.  The borderline reviewers raise several technical questions which the authors have carefully addressed in their rebuttal.

Given that this work makes a new contribution in the area of interactive segmentation and the excellent results presented, the AC recommends that the work be accepted.

**Justification For Why Not Higher Score:**

Some concerns regarding novelty and clarity.

**Justification For Why Not Lower Score:**

Interesting work, first to address the multi-object setting.

---

### Decision · Program_Chairs · 2024-01-16

Accept (poster)